# Sustainable vs. Conventional Approach for Olive Oil Wastewater Management: A Review of the State of the Art

Zakaria Al-Qodah [1,*], Habis Al-Zoubi [2], Banan Hudaib [1], Waid Omar [1], Maede Soleimani [3], Saeid Abu-Romman [4] and Zacharias Frontistis [5]

1    Chemical Engineering Department, Faculty of Engineering Technology, Al-Balqa Applied University, Amman 11183, Jordan; banan.hudaib@bau.edu.jo (B.H.); waid.omar@bau.edu.jo (W.O.)

2    Department of Chemical Engineering, College of Engineering, Al-Hussein Bin Talal University, Ma'an 71111, Jordan; HabisAl-Zoubi@ahu.edu.jo

3    Department of Environmental Health Engineering, School of Health, Qazvin University of Medical Sciences, Qazvin 34197-59811, Iran; Soleimani.m1995@gmail.com

4    Faculty of Agricultural Technology, Al-Balqa Applied University, Salt 19117, Jordan; saeid.aburomman@bau.edu.jo

5    Department of Chemical Engineering, University of Western Macedonia, 50132 Kozani, Greece; zfrontistis@uowm.gr

\*    Correspondence: zak@bau.edu.jo

**Abstract:** The main goal of this review is to collect and analyze the recently published research concerning the conventional and sustainable treatment processes for olive mill wastewater (OMW). In the conventional treatment processes, it is noticed that the main objective is to meet the environmental regulations for remediated wastewater without considering the economical values of its valuable constituents such as polyphenols. These substances have many important environmental values and could be used in many vital applications. Conversely, sustainable treatment processes aim to recover the valuable constituents through different processes and then treat the residual wastewater. Both approaches' operational and design parameters were analyzed to generalize their advantages and possible applications. A valorization-treatment approach for OMW is expected to make it a sustainable resource for ingredients of high economical value that could lead to a profitable business. In addition, inclusion of a recovery process will detoxify the residual OMW, simplify its management treatment, and allow the possible reuse of the vast amounts of processed water. In a nutshell, the proposed approach led to zero waste with a closed water cycle development.

**Keywords:** olive mills wastewater; circular economy; polyphenols; wastewater treatment; agro-industrial wastewater; recovery of valuable ingredients



## 1. Introduction

These days, environmental sustainability should be considered in all industrial and agro-industrial sectors [1]. It can be promoted by producing cleaner products with less waste and by reusing and recovering all valuable components present in wastewater [2]. Some of these industrial sectors generate large amounts of different types of wastewater, such as textile [3], petroleum refining [4], carwash [5], and food [6] industries. Olive oil production processes usually generate large amounts of high load polluted wastewater known as Olive Mill Wastewater (OMW), locally known as "Zibar", [6]. Most of these olive oil-producing processes are found in the Mediterranean countries, where olive trees are planted to produce both olives and olive oil. These countries produce more than 15 M m$^3$/year or about 98% of the world's olive oil production [7,8]. The largest olive oil-producing countries are Spain, Italy, Greece, Turkey, Tunisia, Portugal, Morocco, and Algeria [9]. In addition, olive trees are planted in the Middle East countries, USA, Australia, and Argentina [10].

In Jordan, olive trees are abundant, especially on the country's northern part, where the weather is favorable for this type of tree. Consequently, the olive tree is considered an essential agro-industrial cost-effective contributor to the local economy. However, a relatively large amount of 300,000 m$^3$ of Zibar is produced every season in addition to the production of 120,000 tons of olive mill solid wastes, locally known as "Jift". These produce liquid and solid waste and impose a serious environmental challenge that needs sustainable management [11–14]. Accordingly, this review will focus on the properties of OMW constituents. It will consider the most relevant research published in the last two decades related to OMW treatment using single and combined processes. Then it will compare two main approaches for OMW management, namely; the conventional and the sustainable treatment processes. In the conventional approach, the treatment process's main objective is to treat OMW to meet environmental regulations. On the other hand, the sustainable approach uses a treatment process that compromises the recovery of valuable ingredients found in OMW followed by treatment of the residual wastewater of OMW. In addition, these processes will be analyzed and compared according to their benefits. The work aims to focus more attention on the crucial use of sustainable treatment processes. This integrated–hybrid approach has not been considered before. It is expected to encourage researchers to develop large-scale sustainable treatment processes for OMW to benefit from the valuable constituents of these wastes before or after their treatment.

Accordingly, this review paper uses a systematic methodology with the main objective to compare conventional and sustainable approaches for OMW management and ensure the reliability of the sustainable approach. The first step was to identify all relevant high-quality research works addressing the issue of conventional and sustainable- processes in (OWM) management. Then to select studies that implement appropriate analysis providing a high degree of confidence in the validity, reliability, and applicability of the obtained results. The review is based on collecting a large number of studies published in the last two decades, and then screening them to use all potentially relevant research. After that, a filtering process was applied to exclude the publications from weak journals and pick up only papers from high-impact and high-level journals indexed in web of science and Scopus. The selected relevant, good-quality research papers were classified according to the type of the applied process, and then some of their main results were discussed and compared. From this discussion some conclusions give answers to our main question about the reliability of using sustainable approaches for Olive Oil Wastewater (OWM) management, and structural synthesis is conducted to put the findings together and answer our review question.

## 2. OMW Composition

The composition and characteristics of OMW usually vary depending on many factors, including the geographic location and climate, type, the degree of maturity of olive fruits, the processing procedures, and the method of oil extraction, such as batch or continuous [10]. In addition, the classical processes for olive oil extraction from fruits use continuous processes of two and three-phase centrifugation to affect the composition of OMW, as shown in Table 1 [15].

**Table 1.** Portuguese OMW characterization from 6 olive mills in the campaign. Adapted from Reference [15].

| Factors | OMW I | OMW II | OMW III | OMW IV | OMW V | OMW IV |
|---|---|---|---|---|---|---|
| pH | 6.85 | 5.02 | 4.24 | 5.02 | 4.92 | 5.50 |
| COD (kgm$^{-3}$) | 9.08 | 44.6 | 20.6 | 134.0 | 135.0 | 23.0 |
| BOD (kgm$^{-3}$) | 4.75 | Nd | 11.0 | 40.0 | 42.0 | 7.90 |
| PhC * (kgm$^{-3}$) | 0.03 | 2.54 | 0.61 | 5.40 | 6.16 | 0.25 |
| TS (kgm$^{-3}$) | 7.30 | 33.1 | 15.1 | 117 | 106 | 18.8 |
| VS (kgm$^{-3}$) | 7.10 | 28.4 | 9.80 | 94.3 | 79.2 | 12.9 |

* PhC phenolic compounds, Nd—not determined.

The color of OMW effluent varies from black to dark brown. It is characterized by a high organic load of different compounds, including sugars, lipids, pectin, tannins, polyphenols, and polyalcohols [15,16]. Table 2 shows the details of different components found in untreated and treated OMW.

**Table 2.** Physiochemical characteristics of untreated and treated OMW. Adapted from Reference [17].

| Characteristics | Untreated OMW | Treated OMW |
|---|---|---|
| pH at 25 °C | 5 ± 0.2 | 8.1 ± 0.2 |
| Electrical conductivity (25 °C) (dS m$^{-1}$) | 8.2 ± 0.1 | 14.2 ± 0.1 |
| Chemical oxygen demand (g L$^{-1}$) | 53.3 ± 4.8 | 4.5 ± 0.41 |
| Biochemical oxygen demand (g L$^{-1}$) | 13.42 ± 1.21 | 1.8 ± 0.16 |
| COD/BOD$_5$ | 4 ± 0.72 | 2.5 ± 0.45 |
| Salinity (g L$^{-1}$) | 6.23 ± 0.56 | 12.1 ± 1.1 |
| Water content (g L$^{-1}$) | 960.6 ± 19.2 | 984 ± 19.7 |
| Total solids (g L$^{-1}$) | 39.55 ± 1.98 | 15.9 ± 0.8 |
| Mineral matter (g L$^{-1}$) | 6.5 ± 0.33 | 10.15 ± 0.51 |
| Volatile solid (g L$^{-1}$) | 33 ± 1.65 | 4.8 ± 0.24 |
| Total organic carbon (g L$^{-1}$) | 17.6 ± 0.88 | 3.2 ± 0.16 |
| ortho-diphenols (g L$^{-1}$) | 8.6 ± 0.86 | 0.77 ± 0.08 |
| Total nitrogen Kjeldhal (g L$^{-1}$) | 0.5 ± 0.05 | 0.25 ± 0.03 |
| Carbon/Nitrogen | 35.2 ± 7.04 | 12.8 ± 2.56 |
| Toxicity by LUMIStox (% I B) | 99 ± 9 | 30 ± 3 |
| P (mg L$^{-1}$) | 36 ± 3.6 | 15 ± 1.5 |
| Na (g L$^{-1}$) | 0.8 ± 0.08 | 0.86 ± 0.09 |
| Cl (g L$^{-1}$) | 1.45 ± 0.15 | 1.3 ± 0.13 |
| K (g L$^{-1}$) | 8.6 ± 0.8 | 5.34 ± 0.5 |
| Ca (g L$^{-1}$) | 0.9 ± 0.09 | 3.2 ± 0.3 |
| Fe (mg L$^{-1}$) | 23.4 ± 2.3 | 38.3 ± 3.8 |
| Mg (mg L$^{-1}$) | 186.9 ± 18.7 | 281 ± 28.1 |

It is clear from Table 2 that OMW contains various components, including organic and inorganic compounds or ions. The most important compounds are polyphenols, volatile acids, sugars, polyalcohols, and some nitrogen compounds. The concentration of phenolic compounds could reach 10 g/L, a value that makes OMW toxic with high antibacterial activity [17]. Fortunately, most of these components have economic value. However, even recently, most of the treatment processes for OMW aim to treat the wastewater without getting the benefits of its valuable components.

The annual production rate of OMW effluents in all the Mediterranean region is over $30 \times 10^6$ m$^3$ [18]. This sizeable seasonal production rate of wastewater represents a considerable hazard to the environment due to the high load of pollutants, which needs an intensive management protocol [19]. Unfortunately, the treatment of OMW is still a crucial issue in the Mediterranean region that needs to be resolved due to the severe negative impact on the environment, especially land and water resources. This fact still challenges researchers and industries to present their research findings and technologies to solve OMW problems [20]. Intensive research is needed not just to treat OMW but for clean production of the olive oil industry from one side and the sustainable treatment of the OMW [21].

## 3. Environmental Impact of OMW

The primary waste produced from olive oil extraction processes is OMW. This wastewater had been historically discharged directly onto land, forming a black soil layer near the olive oil mills [22,23]. These days, the annual quantity of OMW effluents produced in the Mediterranean countries is about $30 \times 10^6$ m$^3$ [24]. These effluents have become an increasing environmental challenge due to the significant increase in olive tree planting and the consequent development in olive oil production processes. For example, the shift from conventional mills using pressure into machines using centrifugal force has accom-

panied the production of more significant amounts of OMW [25]. In addition, OMW is considered a complex effluent characterized by a high load of chemical oxygen demand (COD) and various toxic materials [26,27]. Furthermore, OMW is acidic and contains high concentrations of polyphenols, which is mainly responsible for its phytotoxicity and low biodegradability [28]. Accordingly, olive processing is usually associated with many possible adverse effects on the environment, including resource depletion and land degradation, if only classical processes are applied to manage OMW.

The concentration of by-products derived from the olive oil extraction processes differs based on the extraction techniques used. It includes olive oil, which is only about 20% of the overall input volume, and two different waste matrices known as olive cake or olive pomace and OMW wastewater [29,30]. It is well known that the physicochemical characteristics of OMWs are changeable, depending on olive cultivars, climatic conditions, storage time, degree of fruit maturation, and notably the extraction procedure [31]. Table 3 shows the characterization of some Jordanian OMW.

**Table 3.** Characterization of OMW in Jordan. Adapted from References [31–34].

| Property | pH | Phenols | COD | TSS | TDS | Ca | Cu | K | Mg | Na | Zn |
|---|---|---|---|---|---|---|---|---|---|---|---|
| Unit | - | g/L | g/L | g/L | g/L | mg/L | mg/L | g/L | mg/L | mg/L | µg/L |
| Value | 4.6–5.9 | 0.3–3.0 | 10–50 | 5–21 | 2–35 | 2.1 | 0.9 | 0.2–8 | 1.9 | 0.7 | 33″ |

The amount of water used in the mills is usually controlled by labor practices and pressing techniques. Accordingly, the quantity of OMW per kg of oil produced and the concentration of its constituents, shown in Table 3, usually vary from country to country and from mill to mill. The high concentration of toxic polyphenols in addition to the high organic COD load suggested that the uncontrolled release of OMW on lands and water resources or stored in open ponds, will present an emerging polluting power. This will leads to the spreading of foul-smelling materials in air and poisonous materials in the soil as well as water. In addition, OMW has bad odor caused by different low-boiling organic matters and volatile acids. Moreover, due to anaerobic conditions, some gases are usually released from the reservoirs, such as methane and other potent acid gases like hydrogen sulfide, which lead mainly to air pollution [34].

It is well known that OMW is toxic to both plants and microorganisms, including soil microflora [35,36]. The high salt content and the relatively low pH of OMW might be phytotoxic to soil biological properties [37,38]. The OMW toxicity is essentially due to the presence of some monomeric phenols [31]. Severe phytotoxic effects may occur on higher plants, mainly during germination and seedling development, due to the enhancing action of phenolic compounds on seed dormancy [39]. Moreover, the C/N ratio of OMW has an adverse effect on the biodegradation and humification processes [16]. Soil properties, seed development, and plant growth were evaluated in response to three different types of olive mill wastewater: untreated olive mill wastewater (UOMW), treated (TOMW), and bio augmented olive mill wastewater (BOMW). The results showed that despite the high initial OMW acidity, the OMW application reduced soil pH by 0.2 units in six months follow-up periods. Comparably, OMW application also increased the soil electrical conductivity, EC. The study also compared seeds development using different concentrations of UOMW and TOMW applied for various species and showed that the growth was highly inhibited for all examined samples when UOMW/water ratio was lower than 1/10. However, OMW can work as a high nutrient soil fertilizer if a suitable separation process removes the toxic materials.

Many other authors reported adverse effects from OMW application on cultivated plants. These negative impacts resulted from OMW application close to the sowing period. For example, Boz et al. [40] performed field experiments that stated that the adverse effects on the wheat were confined to the primary stages and not detected at the late production stages. Casa et al. [41] reported that undiluted OMW has completely inhibited durum

wheat seed development. El Hadrami et al. [42] also noticed significant seed germination reduction after spreading OMW of various concentrations or OMW phenolic extracts in tomato, chickpea maize, and durum wheat. Furthermore, Quaratino et al. [43] observed maize seed development reduction with increased OMW volumes in loamy sand soil. Moreover, Greco et al. [38] noticed a phenolic dose-dependent phytotoxic effect on the germination of tomato and English cress seeds.

Based on the above observations, OMW represents a severe environmental challenge if it is illegally poured on lands and the open environment without treatment. Accordingly, there is a necessity for efficient, sustainable plans for OMW management to reduce their adverse environmental impact from one side and to develop sustainable processes to recover their valuable constituents and use them in many vital human and agricultural applications.

## 4. The Conventional Treatment Processes of OMW

This section will discuss and compare the results of several classical treatment processes used to treat OMW. These processes include physical, chemical, and biological technologies and with various combinations. It should be noted that the main objective of these conventional treatment processes is to treat OMW and to obtain remediated wastewater that can be used in agriculture or other non-drinking uses.

### 4.1. Biological Based Processes

The application of biological processes for wastewater treatment is usually verified as a reliable, environmentally friendly, and economical approach. In the present case, the biological treatment of OMW can achieve considerable success in the effective removal of organic matter, especially phenolic compounds and other substances. However, particular microorganisms must be selected for the bio-digestion of OMW since phenolic compounds could be inhibitory to most microorganisms.

Hamdi et al. [44] applied an aerobic detoxification step for OMW, followed by a methanization and a final aerobic post-treatment step. The first step was supplemented with sulfate and ammonium and was carried out by the growth of *Aspergillus niger* in a bubble column. *A. niger* as a wild strain was able to decrease OMW toxicity and increase its biodegradability by degrading the phenolic compounds. The COD removal efficiency was about 58% with the production of biomass containing 30% proteins ($w/w$).

Gunay and Karadag [45] review recent developments in the anaerobic digestion of OMW. The co-digestion of OMW with different wastewater sources such as manures, slaughterhouses, and whey was discussed. Furthermore, a co-digestion of OMW with sludge and microalgae was reviewed. Co-digestion improved methane yield by making a balance between nutrient and alkalinity levels. Additionally, a broad evaluation of research concerning pretreatment has been carried out by assessing their performances. Ehaliotis et al. [46] investigated the aerobic biological treatment of OMW using *Azotobacter vinelandii* strain. They focused on this microorganism adaptation and population dynamics and its capacity to fix nitrogen and generate fertilizer from OMW. Their results showed an initial phase of physiological adaptation. The existence of phenolic compounds restricts the growth of *A. vinelandii*, but motivates nitrogen fixation, and subsequently attains a fast growth phase as a decay in phytotoxicity OMW occurs [47].

Application of ultrasound energy in the pretreatment process of OMW in order to enhance the degradation of biomass was reported by Oz and Uzun [47] and Al-Qodah et al. [18]. Preceding the anaerobic batch reactor digestion step, low frequency ultrasound irradiation was applied on the OMW, and its potential as pretreatment step was evaluated. Experiments were conducted to find out the optimum conditions of ultrasonic applications. Al-Qodah et al. [18] concluded that the best results were achieved by introducing ultrasound of frequency 20 kHz to diluted OMW with the intensity of 0.4 W/mL for 10 min. According to Oz and Uzun [47], the main role of ultrasound induced pretreatment is the improving of solubilization of the organic constituents in the OMMW wastewater to facilitate anaerobic digestion. The evaluation of the degree of organic matter solubilization

was carried out by calculating the ratio of the demand for soluble chemical oxygen to the demand for the total chemical oxygen. This ratio was measured before and after ultrasonic irradiation. This ratio was increased from 0.59 to 0.79 due to acoustic cavitation under the above-mentioned optimum conditions. This was reflected in the amount of biogas and methane generated during the anaerobic digestion. It has been experimentally proved that an increase of 20% in methane and biogas production was measured when feeding the batch reactor for an anaerobic digestion process with pretreated OMW with ultrasound [48].

In another study, Al-Qodah et al. [18] applied ultrasound waves to reduce phenolic compounds and other organic pollutants from OMW. The adopted ultrasonic irradiation was combined with aerobic biodegradation as the treatment technique. The researchers examined various operational parameters such as the duration of ultrasonic irradiation, power intensity, and frequency to determine their effects on polyphenol and COD degradation. They found that after 90 min of continuous exposure of OMW to ultrasonic irradiation at 25 °C, COD value was not affected, and 81% degradation of the total phenol was achieved. However, COD removal efficiency was about 80% in the aerobic biological treatment process. They concluded that acoustic cavitation degraded polyphenols and thereby significantly enhanced the biological step's efficiency [18].

Chiavola et al. [24] studied the removal of biodegradable organic matter found in OMW using sequenced batch reactors. This process uses OMW, which is first sieved and then diluted with tap water. The dilution fractions studied (OMW: tap water, $v/v$) were: 1/25, 1/32, 1/16, and 1/10. Results revealed that the biodegradable organic compounds were utterly eliminated for each significant pollutant studied, with average efficiencies of 90% for COD and 60% for total polyphenols. The pretreatment or postreatment step using membrane processes such as: reverse osmosis (RO), nanofiltration (NF), and ultrafiltration (UF) was also investigated. The downstream membrane processes permitted attaining treated water with excellent electrical conductivity, pH, and COD. However, the concentration level of total phenols has not attained the required standards to be reused [23].

González and Cuadros [48] applied an aerobic pretreatment step prior to anaerobic digestion to eliminate phenolic constituents of OMW and also enhanced COD reduction. The results showed a decline of 21% of COD and 90% of polyphenols concentration for an aeration time of seven days. In addition, the results revealed that when the aeration time of the pretreated OMW was five days, the optimum amount of produced methane was 0.39 m$^3$ for each kilogram of removed COD. This yield value was about 2.3 times greater than that obtained from non-pretreated OMW [48].

The overall energy balance was considered an essential parameter when choosing a certain OMW pretreatment process [49]. Therefore, the cost of energy consumption should be compared with biogas production. Numerous pretreatment processes were analyzed by evaluating their potential of biochemical methane production and energy sustainability index. Results revealed that the most optimum and practical pretreatment strategy was using calcium carbonate in the pretreatment. The estimated biogas production was equal to 21.6 NL/L, and the index of energy sustainability was 14, which indicates that the energy gained from methane production is 14 times the total energy consumed [49].

In another work [50], OMW was detoxified, employing a wild strain of *Candida oleophila* isolated from OMW. Germination measurements were used to evaluate the treatment's potential by incubation with isolates of the wild strain of *C. oleophila* to remove specific pollutants such as the organic constituents and polyphenolic compounds. It was found that about 50% removal of organic matter and nearly 83% reduction in the concentration of polyphenolic compounds was achieved. Moreover, a measured 50% reduction in antimicrobial activity of OMW was reported. This indicated that treating the OMW with the *C. oleophila* isolate has been proven to be an effective and promising treatment method for OMW reclamation [50].

The application of biological methods faces some difficulties due to the presence of phenolic compounds, which act as inhibitory to the used microorganisms. Using anaerobic bacteria has some disadvantage due to their lower growth rates, whereas aerobic

microorganisms exhibit higher growth rates. On the other hand, anaerobic digestion has the advantage of low energy costs, and there is little production of sludge, and there is also the possibility of energy recovery due to production of methane in the last stages. However, aerobic biological treatment is applied as a pretreatment step to remove the toxic phenols prior to applying the main treatment step.

*4.2. Physio-Chemical Treatments*

Sarika et al. [50] investigated the pre-treatment of OMW by flocculation using cationic and anionic polyelectrolytes. They reported that a significant reduction in COD and complete removal of the TSS was achieved with most of the tested flocculants. The minimum dosage of the flocculants to achieve solid-liquid separation ranged from 2.5 to 3 g/L. Based on these results, a post-treatment process of the liquid phase was proposed, for instance, biologically based methods, applying high power ultrasounds, advanced oxidation process, or a hybrid combined system. On the other hand, the separated solid phase can be used as composted solid agricultural wastes to produce fertilizers [51].

A two sequential process was investigated in detail by Stoller [52] where membrane technology was used after the coagulation process. This combined system was used to reduce the COD load to acceptable levels of less than 500 mg/L. This value allows the treated water to be legally discharged to the municipal sewer. In the experiments conducted, the treated water by coagulation using $Al_2(SO_4)_3$ as a coagulant was subjected to four successive batch membrane processes: MF followed by UF process and then NF membrane and finally RO process. A schematic diagram of the pilot plant used in the investigation is shown in Figure 1. The coagulation process and the type of the coagulant were found to affect the membrane fouling significantly [52].

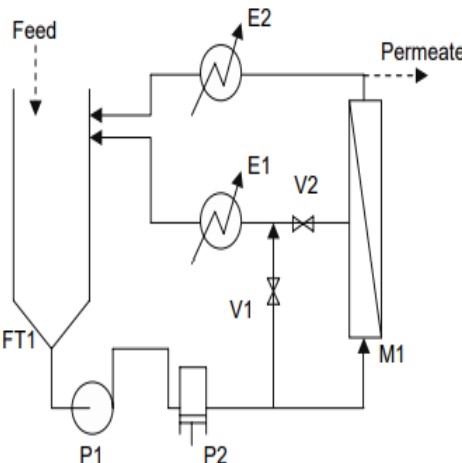

**Figure 1.** Schematic diagram of the membrane separation system used after the coagulation. Reprinted with permission from Reference [52]. Copyright 2005 Elsevier.

From the previous investigations, it can be concluded that coagulation-flocculation is an effective pretreatment process. However, further studies have also investigated other options for conventional chemicals. For example, Biopolymers such as chitosan were used [53], or other wastes generated from different industries are considered new resources. Fragoso and Duarte [54], investigated the use of sludge produced from drinking water treatment plants as a replacement for the classical coagulation/flocculation materials. The results indicated that COD and total phenols concentration reduction was about 50%. On the other hand, the reduction in the total suspended solids and total solids was 70% and 45%, respectively.

El-Gohary et al. [55] oxidized diluted OMW using Fenton reaction and then fed it to a two-staged anaerobic sludge blanket reactor. The treatment process is described clearly in the block diagram shown in Figure 2.

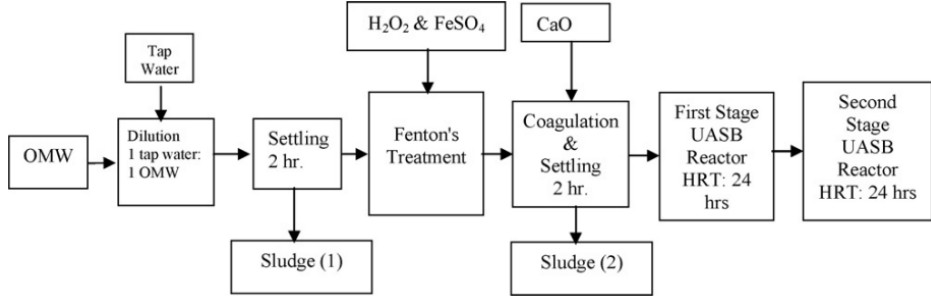

**Figure 2.** Treatment of OMW using the combined Fenton catalytic oxidation process and the anaerobic sludge two-staged blanket reactor. Reprinted with permission from Reference [55]. Copyright 2012 Elsevier.

The concentration of phenols before and after this treatment for different processes are shown in Table 4.

**Table 4.** The concentration of the polyphenolic compounds for both raw and treated OMW Adapted from Reference [55].

| Compound | R. Time | Raw | | | Treated Effluent | | |
| --- | --- | --- | --- | --- | --- | --- | --- |
| | | F1 | F2 | F3 | F1 | F2 | F3 |
| Benzoic acid | 3.307 | 45.0 | 60 | ND | 0.12 | ND | 5.6 |
| Salicylic acid | 3.518 | 162.5 | 160.9 | 468.5 | 16.9 | 6.4 | 67.4 |
| 2,4 Dihydroxy benzoic acid | 4.300 | 426.5 | 426.8 | 126.0 | ND | ND | ND |
| Tyrosol | 4.849 | 83.8 | 164.7 | 38.4 | ND | ND | ND |
| 2,4 Dyhydroxy bnzaldehyde | 6.38 | 368.7 | 1048 | ND | ND | ND | ND |
| Syringic acid | 7.55 | 119.7 | 610.4 | ND | ND | ND | ND |
| 4-Hydroxy benzoic acid | 9.274 | 426.7 | 120.1 | ND | ND | ND | ND |
| 2,4 Dihydroxy cinnamic acid | 10.499 | ND | 96.1 | ND | ND | ND | ND |
| Syringaldehyde | 13.350 | ND | 354.2 | ND | ND | ND | ND |
| p-Coumaric acid | 17.631 | ND | 466.6 | ND | ND | ND | ND |
| Ferulic acid | 18.087 | ND | 278.5 | ND | ND | ND | ND |

F1 = free phenols, F2 = phenols from esters, F3 = phenols from glycoside compounds. ND—not detected.

After the Fenton reaction, the generated sludge from the oxidation of OMW can be eliminated by developing an efficient solid, liquid separation process such as flocculation using different flocculants. Martinez Nieto et al. [56] tested the potential of four types of flocculants for sludge separation after the OMW oxidation step in a system described by the schematic diagram displayed in Figure 3. They found that 6 mg/dm$^{-3}$ of Nalco 77,171 was the optimum flocculants dose. Nalco 77,171 received 13.5% sludge separation, and 86.5% clarified treated water [56].

A similar approach was applied by Alver et al. [56] using Fenton oxidation of phenolic pollutants. They reported that the optimum conditions were pH 3, ferrous ion concentration equal to 2.5 g/L, while the ratio of the ferrous ion to hydrogen peroxide was 2.5. At these optimum conditions, the COD reduction was 65%, and that of phenolic compounds was 87.2% [57].

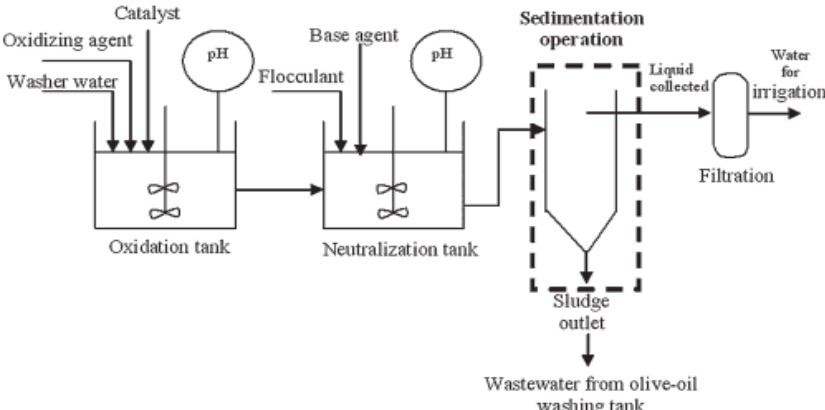

**Figure 3.** Sludge separation after OMW oxidation process. Reprinted with permission from Reference [56]. Copyright 2008. Wiley and Sons.

*4.3. Electrochemical Treatment Processes*

Numerous electrochemical processes have previously been used for the reclamation of OMW. Among these processes are electro-Fenton and electrocoagulation. In addition, electrochemical oxidation using polyaluminum chloride and conductive diamond were applied. Moreover, the in situ production of active chlorine and bulk electrolysis with different electrodes were used to treat OMW.

OMW reuse for agricultural irrigation using anaerobic digestion after the electro-Fenton oxidation was investigated by Khoufi et al. [58]. Their study revealed that the electro-Fenton processes reduced the total phenolic by 65.8% and the toxicity by 33.1%. In addition, the application of electrocoagulation after anaerobic digestion can lead to a complete detoxification of OMW.

There are a few advantages of the electrocoagulation, such as treatment without applying any chemical reagents, and the process itself is easy for management and automation. In addition, the consumption of electric power is low enough to make the process economic with efficient COD reduction.

Tezcan Un et al. [59] treated OMW by coagulation using the coagulant polyaluminum chloride combined with an electrochemical process with the addition of $H_2O_2$. The potential of the combined process for the treatment of OMW was optimized by examining the parameters affecting the treatment efficiency such as electrode type, current density, amount of $H_2O_2$ and coagulant dose. They found that the COD removal efficiency ranged from 62% to 86%, whereas a 100% removal of both turbidity and grease oil was achieved. The optimum current density was between 20 and 75 mA/cm$^2$ depending on the amount of $H_2O_2$ used [60].

In another study, Tezcan Un et al. [61] investigated an electrochemical treatment process for OMW without pre- or post-treatment. They used the electrochemical process shown in Figure 4 to study the parameters affecting COD, phenols, grease oil, turbidity removal, and the specific energy consumption, such as rate of recirculation, the concentration of NaCl, temperature, and applied current density.

The results of Tezcan et al. [60] revealed that the consumption rate of specific energy was ranged between 5.35 and 27.02 kWh/(kg COD). They found that specific energy consumption decreased as sodium chloride concentrations, circulation rate, and temperature increased. On the other hand, the efficiency increased as the current density increased. This process was very efficient and reached about 99.6, 99.85, 99.85, and 100% removal of COD, turbidity, oil grease, and phenol [60].

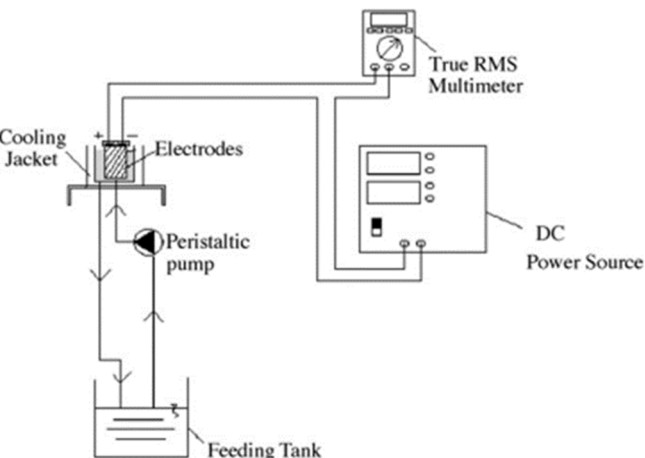

**Figure 4.** The electrochemical process used for the treatment of OMW. Reprinted with permission from Reference [60]. Copyright 2008. Elsevier.

Cañizares et al. [61] investigated Fenton oxidation, ozonation, and conductive diamond electrochemical oxidation processes on the treatment of simulated wastewater contaminated with different organic pollutants and real OMW. According to their results, the electrooxidation using conductive diamond showed the optimal results for eliminating all pollutant types. However, the efficiency of ozonation and Fenton oxidation was strongly influenced by the pollutant type [61].

Electrochemical oxidation of OMW was studied using bulk electrolysis and cyclic voltammetry methods by Papastefanakis et al. [62]. The authors used DSA Ti/RuO$_2$ anodes and a fixed temperature of 80 °C in an acidic medium. The studied parameters were potential windows, current density, and sodium chloride concentrations. The experimental results revealed that electrochemical oxidation leads to 52, 38, 84, and 86% removal of COD, total organic carbon (TOC), phenols, and color, respectively [63].

The selection of anodes in electrochemical oxidation needs to be carefully studied. Anodes should meet certain requirements such as good conductivity, high specific surface area, excellent adsorption capability, better catalytic and electric capabilities to achieve a higher degradation ability. An optimum anode selection should fulfill the low cost, high chemical stability and excellent electro catalytic activity in addition to avoiding the drawback of low oxygen evolution over potential and the higher cost material.

### 4.4. Photocatalytic Degradation Methods

Among different advanced oxidation processes, photocatalytic degradation has widely been applied to purify different polluted waters and wastewaters, to eliminate many inorganic contaminants and disinfection [53]. Light-assisted AOPs (O$_3$/UV, H$_2$O/UV) reduced the total phenols and COD by more than 99% [63,64]. One application of photocatalytic processes in the treatment of OMW was performed by Chatzisymeon et al. [65] in a laboratory-scale batch photo-reactor irradiated by 400 W Ultraviolet (UV-A) rays using titanium dioxide as the catalyst. They tested the influence of most operating parameters such as organic matter initial concentration, pH, and treatment time. The COD removal was found to be proportional to the treatment time and its initial loading. Based on their results, the researchers concluded that the OMW photocatalytic treatment is an outstanding process for OMW reclamation.

Reduction in OMW organic matter and toxicity using fungi biological treatment followed by photo Fenton oxidation was studied by Justino et al. [66]. Dilutes samples of OMW were treated by the wild fungi *Pleurotus sajor caju*. Fungi treatment reduced phenols and COD by 77% and 72.91%, respectively. Subsequently, the photo Fenton process was applied and found to be effective in color removal. Another operation mode was applied, in which the photo Fenton process was followed by biological treatment. Photo Fenton photo

process showed a high capability to treat non-diluted OMW samples and achieved 100, 76, and 92% removal of organic matter, COD, and phenolic compounds, respectively [66].

Recently, Ochando-Pulido et al. [67] conducted a laboratory-scale study to investigate the photocatalytic degradation of OMW. The main technical-economical obstacle was the difficulty of catalyst recovery. The problem was solved using a developed and advanced Nano-sized photo-catalyst with ferromagnetic characteristics. The photocatalyst resulted in 58.3% COD reduction, 27.5% phenols elimination of 27.5% and 25% TSS removal. On the other hand, when a pretreatment process of pH and temperature-controlled flocculation was performed, up to a 91% increase in the overall COD elimination effectiveness was attained. Based on the previous results, the photocatalytic degradation technology showed promising potential as a treatment process with outstanding capabilities in OMW reclamation [67].

Ruzmanova et al. [68] investigated OMW purification by photo-catalysis using Nitrogen-Doped $TiO_2$ Nanocrystals prepared by the sol-gel method [69–73]. The modified materials showed higher potential than other studies with unmodified catalysts, attaining a COD reduction greater than 60%. The modified titanium dioxide was very reactive to solar light and might characterize as a very encouraging process for reducing the organic load in OMW.

Papaphilippou et al. [73] recommended a combined process for OMW treatment involving a consecutive of coagulation-flocculation followed by extraction of phenols and then Photo Fenton processes. The treated water produced after Photo Fenton has a chemical oxygen demand elimination of about 73% and a reduction in phenolic compounds up to 87%. Recently, Malvis et al. [74] investigated an integral process based on flocculation, photolysis and microfiltration followed by micro algal for the treatment of OMW as shown in Figure 5.

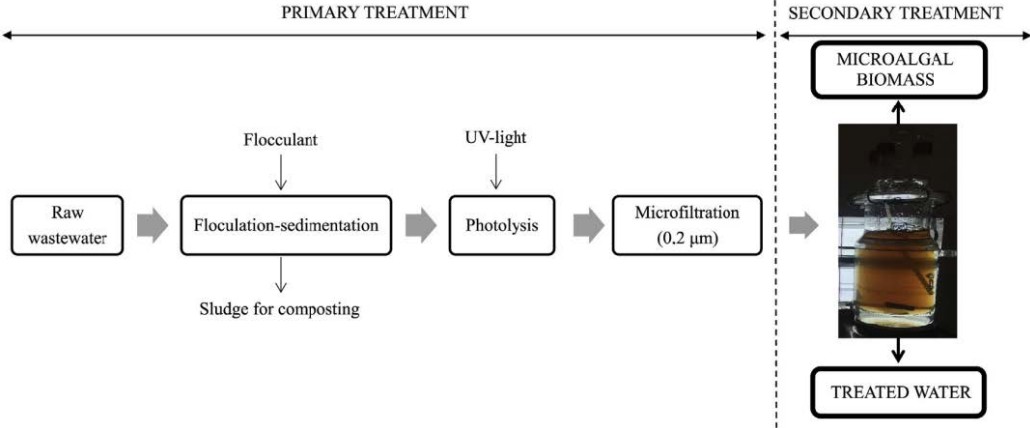

**Figure 5.** Integrated process for OMW treatment and the generation of high added value algal biomass. Reprinted with permission from Reference [74]. Copyright 2019. Elsevier.

According to their results, the COD reduction was 57.5%, 88.8%, and 20.5% by the flocculation, photolysis and microfiltration, respectively [74].

A successful design of OMW treatment by photo-catalysis is the energy consumption per unit mass of pollutant removed, which is dependent on the L influent COD. Furthermore, monitoring toxicity during photocatalytic treatment and the process must guarantee that OMW was almost completely detoxified. The toxicity of the rerated OMW is dependent on the influent COD loadings.

### 4.5. Advanced Oxidation Processes

Several studies demonstrated the removal of different recalcitrant or toxic pollutants by advanced oxidation processes (AOPs) [75]. This group of processes is based on the creation of strong oxidizing agents such as hydroxyl radical (*OH), which oxidize the most unselective pollutants and enhance the biodegradability. Moreover, a coagulation process based on iron besides $H_2O_2$ similar to the Fenton process was studied. This combination

proved to be an efficient OMW pre-treatment process that enhances the degradation of organic constituents. As a result of the acidity of the OMW and the substantial effectiveness in eliminating phenols, Fenton and Photo-Fenton treatment methods were considered as suitable processes for treating OMW.

Fenton's process relies on the generation of hydroxyl radicals using catalytic activation of hydrogen peroxide stimulated by iron ions according to Equation (1) [76]:

$$Fe^{2+} + H_2O_2 \rightarrow Fe^{3+} + HO^- + HO^o \tag{1}$$

Therefore, this process could be used for different applications since it can occur at room temperature and atmospheric pressure [77] while no advanced equipment is required [77,78]. Furthermore, there are no safety or environmental impacts concerned with hydrogen peroxide [79]. The main disadvantage of this technology is the separation of the dissolved iron from the treated OMW. The "photo-Fenton" is either homogeneous or heterogeneous and can be combined with semiconductor photo catalysis using UV active catalysts ($ZrO_2$ and $TiO_2$, and mixed Al–Fe pillared clays, named FAZA). Under the optimum conditions, Badawy et al. [80] showed that the photo-Fenton treatment could attain a percentage reduction in total suspended solids (TSSs), COD, total phenolic compounds (TPC), and TOC of 98.31%, 87%, 97.44%, and 84%, respectively.

Pham Minh et al. [80] investigated OMW oxidation by the wet air catalytic process with ruthenium and platinum reinforced with zirconium or titanium, combined with anaerobic digestion. The results showed that the "total organic carbon" (TOC) was effectively removed about 97% and almost eliminated the phenolic content in catalytic wet air oxidation at 70 bar pressure and 190 °C temperature. Furthermore, the phytotoxicity of the treated OMW effluent was reduced. The ruthenium catalyst was proved to be steady over an elongated operational period. A high degree of mineralization was attained, accompanied by an improvement in the produced methane yield, which uses the anaerobic digestion method. Nevertheless, experiments were performed on real OMW, diluted two times [81].

Subsequently, Azabou et al. [82] studied a catalytic oxidation process to OMW reclamation using wet $H_2O_2$ in a compacted process. Numerous biological treatment methods follow the oxidation process. These processes were conducted by using the heterogeneous catalyst "montmorillonite-based aluminum-iron pillared interlayered clay [(Al–Fe) PILC]". Furthermore, the ratio of [(Al–Fe) PILC] to $H_2O_2$ was tested under atmospheric pressure with ultraviolet irradiations and at 25 °C or 50 °C. The results showed that the photocatalytic process did not affect the raw OMW stream. However, a significant decrease in color, total phenol concentrations, and COD was achieved through the late process [82].

Another interesting study investigated an advanced process using the Fenton reaction to degrade the organic materials present in OMW streams produced from the two phase olive oil factory [83]. This study tested numerous laboratory-scale methods to implement the inexpensive $Fe^{3+}$ salts instead of $Fe^{2+}$ salts. The study investigated the potential of "Mohr salt [$(NH_4)_2Fe\,(SO_4)_2\,6H_2O$]" catalyst, "ferric perchlorate", $Fe(ClO_4)_3$ and "ferric chloride", $FeCl_3$ in addition to the optimum ratio of catalyst to oxidant and operational conditions. The result revealed that the organic material was effectively degraded utilizing $FeCl_3$ as a catalyst with $H_2O_2$. The removal effectiveness of organic content and phenolic constituent was above 95%. Additionally, ferric ions aided in avoiding the depletion of the oxidant $H_2O_2$ when $Fe^{2+}$ ions are transformed into $Fe^{3+}$ ones, which occurs in non-productive parallel reactions. These results proved that a Fenton-like reaction is a comparatively low-cost solution for the OMW treatments. The reclaimed water from this oxidation process was used to irrigate the farms directly for agricultural purposes. In additional research, the OMW treatment was optimized by a Fenton-like process on a pilot scale using a continuously mixed tank reactor [84]. Measurements showed that a steady stat for Fenton reaction was reached within 3 h. The oxidation process of organic constituents in OMW was significantly dependent on the pH. The characteristics of the pilot plant effluent were: COD (129 mg/L) and total phenols (0.5 mg $L^{-1}$). The treated

water was suitable for agriculture to irrigate the farms or could be directly pumped into the municipal systems for wastewater treatment [84].

Experiments were conducted by Martinez Nieto et al. [85] on an industrial scale plant for OMW treatment using the Fenton reaction. The experiments were conducted with a large-scale production capacity ranging from three to five cubic meters per hour. Ferric chloride was used as the catalyst to activate $H_2O_2$. The COD of the inlet feed OMW was about 2684 mg/L, whereas the treated water has a COD of approximately 371 mg/L. Therefore, more than 86% removal of COD was achieved. The treated water meets the standards to be used for irrigation purposes or it can be treated with domestic wastewaters [84]. Table 5 summarized different oxidation techniques used for OMW treatment [86].

**Table 5.** Summary of oxidation techniques for OMW treatment. Adapted from Reference [86].

| Oxidation Technique | Conditions | Removal Results | Ref. |
|---|---|---|---|
| Fenton using $FeCl_3$ | Initial conc. of $H_2O_2$ 5% $w/v$, $[FeCl_3]/[H_2O_2]$ = 0.1 $w/w$, Reactor Volume = 3 $dm^3$, agitation speed = 60 rpm, operating time = 180 min, pH = 3, and temperature =298 K | 99.8% of total phenols | [55] |
| AOPs ($O_3/UV$, $H_2O_2/UV$) | pH values of 2, 7 and 9 for varying $H_2O_2$ dosages between 250 and 1000 mg $L^{-1}$ at 20 °C | 99% of total phenol | [63,64] |
| Photo-Fenton | Temp. (20 °C), stirring (100 rpm), Addition of Iron sulfate heptahydrate (7.5 mL; 0.5 mol $L^{-1}$), pH = 4.0, aliquots of $H_2O_2$ (30% $v/v$), UV lamp (VL-6-LC, 245–365 nm) | 100% of phenols | [87] |
| Photo-Fenton | catalysts used: ($TiO_2$-A), ($ZrO_2$) and FAZA, cylindrical photo reactor (0.85 L), UV mercury lamp the UV emitter range from 100 to 280 nm wavelength | 97.44% total phenols | [80] |
| Wet $H_2O_2$ photocatalytic oxidation | at 298 K. Irradiation 30 W UV-lamp, 0.5 g $L^{-1}$ of the catalyst, $H_2O_2$ added, $2 \times 10^{-2}$ M., catalyst: 0.5 g $L^{-1}$ (Al–Fe)PILC | 86% of caffeic acid and 70% Hydroxytyrosol | [88] |
| Ozonation | ozone concentrations varying from 10 to 70 mg $L^{-1}$, at ambient temperature, the initial COD 1100 and 44,000 mg $L^{-1}$ | 80% phenol | [89] |

### 4.6. Combined-Integrated Treatment Processes for OMW

Many studies examined combined systems or integrated processes where different treatment methods can be used in sequenced paths to achieve treated wastewaters that are environmentally acceptable and fulfill the required standards to be released to water bodies. A summary of some recent and promising combined treatment methods and their efficiency are listed in Table 6.

**Table 6.** A summary of some recent combined treatment methods and their achieved level of purification.

| Combined Process | Conditions | Purification Achieved | Ref. |
|---|---|---|---|
| Wet $H_2O_2$ catalytic oxidation followed by up-flow anaerobic sludge blanket (UASB) reactor | Initial COD = 23,400 mg $O_2$ $l^{-1}$, Initial TOC = 4250.3 mg $L^{-1}$, Time of reaction = 90 min, pH 3, $Fe^{2+}$:$H_2O_2$ ratio = 1:10, $FeSO_4 \cdot 7H_2O$ = 14 g $L^{-1}$, $H_2O_2$ = 34.3 g $L^{-1}$, dilution ratio = 1:4 | Reduction in COD, $BOD_5$, TOC and TP by 77%, 78%, 71% and 61%, respectively. | [55] |
| Acid cracking and granular activated carbon adsorption followed by biological process. | optimal contact time (24 h), ptimal GAC dosage (20 g/L), pH value to less than 2, Volumetric Exchange Ratio (0.2), temperature (20 ± 2 °C) | 90% and 76% removal efficiency of COD and TPP, | [90] |

**Table 6.** *Cont.*

| Combined Process | Conditions | Purification Achieved | Ref. |
|---|---|---|---|
| Photocatalytic and membrane processes | Batch photoreactor containing a catalytic ($TiO_2$) membrane, concentration of $H_2O_2$ in the photoreactor was (5 mM), OMW diluted 1:100 $v/v$, 24 h, high pressure UV lamp (450 W) | Reduction in phenol and COD by 90% and 46–51%, respectively. | [91] |
| Coagulation-flocculation then extraction of phenols, followed solar photo-Fenton. | coagulant: (6.67 g/L $FeSO_4 \cdot 7H_2O$), flocculant: (0.287 g/L FLOCAN 23), extraction for 15 min with ethyl acetate at a solvent to sample ratio of 2:1 ($v/v$), oxidation for 240 min at 0.2 g/L $Fe^{2+}$, 5 g/L $H_2O_2$ and pH = 3 | 73% of COD removal | [73] |
| Ultrasonic irradiaton combined with biodegradation | sonochemical degradation at power intensity: 7 $W/cm^2$; volume: 700 mL; pH: natural; temperature: $25 \pm 1$ °C, ultrasound frequency: 351 kHz & 206 kHz. | 80% COD removal efficiency | [18] |
| Fenton and anaerobic biological process | fixed $H_2O_2$/COD ratio of 0.20, pH = 3.5 and a $H_2O_2/Fe^{2+}$ molar ratio of 15:1, microorganisms immobilized in Sepiolite | 64 to 88% COD reduction, and generation of 281 to 322 $cm^3$ of $CH_4$/g COD removed. | [92] |
| Ozonation and EC processes | Electrocoagulation: 45 $mA/cm^2$ after 70 min by using coupled iron–aluminum electrodes, pH = 6, ultraviolet radiation at 253 nm. | 96.4% COD reduction | [93] |
| Modified surfactant (L167-4S) and a cataionic hydrotropes | Surfactant: sodium polypropylene oxides sulfate (L167-4S), combined with cationic hydrotropes tetra butyl ammonium bromide (TBAB) 1:2 molar ratios. | Phenols recovery achieved was in the range of 99.5–99.8%. | [94] |
| Acid flocculation followed by photocatalytic membrane | Membrane UF, Pretreatment processes Acid fluculation and AF and photo catalysis PC, $J_b$ [9.4 L $h^{-1}$ $m^{-2}$], $TMP_b$ [9 bar] $\propto$ [0.0110 L $h^{-2}$ $m^{-2}$ $bar^{-1}$] | Increase the plant productivity by 18–59%. | [20] |
| Combined Ozone/Fenton Process | constant $H_2O_2/Fe^{2+}$ molar ratio of 10, ozonation time (60–120 min). | Reduction in color, DOC and BODs by 21%, 49% and 22%, respectively. | [95] |
| Coagulation-flocculation using 0.1 g of $\gamma$-$Fe_2O_3$ nanoparticles with 1 kg sand | ferric nanoparticles used with sand in a ratio of 0.1 g $Fe_2O_3$ and 1 kg of sand, dilution rate for OMW with tap water is (1:0.5), quantities for ferric chloride and lime are (1:0.5) mg $L^{-1}$ | Reduction in total phenols, BOD5 and COD of 99%, 95.3% and 97.2%, respectively. | [32] |
| Liquid phase extraction, then membrane separation surfactant enhanced aquifer remediation. | solvent (pure water and 50% E-50% W), optimum TPC extracted values were achieved using 40 g of two-phase olive oil and 100 mL solvent, without the addition of HCl, at room temperature 25 °C, after 1 h stirring at 100 rpm. | Reduction in the concentrations of phenols, COD and carbohydrates to 10 mg/L, 284 mg/L and 146 mg/L, respectively. | [96] |
| An integration of applying coagulation method as pre-treatment followed by biological processes. | different levels of pH (4.5, 4.0 and 3.0), constant dosage of two coagulants—aluminum sulfate (Alum) and chitosan equal to 400 mg/L and 100 mg/L, respectively. coagulant dosage, ranging from 400 to 1200 mg/L of alum and 300 to 700 mg/L of chitosan, 60 min of sedimentation. | The removal in the coagulation and biological processes of phenols, COD and TOC were 62.89% and 99.6%; 57.16% and 82.5%; 16.76% and 71.9%, respectively. | [1] |
| Stage 1. "Ozonation, primary sediment separation, Electrocoagulation and lime suspension treatment". Stage 2. "Ozonation, oxidation, and adsorption on charcoal". | Prepared catalyst ($O_3$/($Fe_2O_3$ + CuO)/Clay), pre-ozonated raw OMW has been exposed to electrolytic treatment using a cell supplied with four iron plates as electrodes, lime $Ca(OH)_2$ as suspension was added to adjust the pH to a value of (11–11.5) | Stage 1—59, 86, 70, and 91% of COD, total phenols, color and TS reduction. Stage 2—96% of COD, 100% of total phenols and color and TSS reduction | [97] |

**Table 6.** *Cont.*

| Combined Process | Conditions | Purification Achieved | Ref. |
|---|---|---|---|
| Coagulation-flocculation, and oxidation with $H_2O_2$. | coagulation-flocculation's optimal conditions are 1.5 g/L of $Al_2(SO_4)_3$ and 3 g/L of quicklime and a equal to pH = 8.5, hydrogen peroxide at a rate of 30 g/L. | The value of COD removal can reach 91% | [98] |
| Natural flotation followed by anaerobic-aerobic biodegradation | aeration at a rate of 3.5 L/min, Aeration was maintained discontinuously (every 30 min) Phytotoxicity test are conducted in the darkness at room temperature (25 °C) for 120 hr of exposure. | Reduction in turbidity, COD, polyphenol, nitrate, ammonium and phosphorus by 67.5%, 29.1%, 25.2%, 93.9%, 77.1% and 81.8%, respectively. | [99] |
| Infiltration percolation in column followed by aerobic biological treatment | Infiltration using granular activated carbon column mixed with 15% of lime, biological treatment under aerobic conditions. The total volume treated is 1500 mL diluted 15 times with distilled water and neutralized with $H_2SO_4$ (0.1 N) | Reduction in COD, $BOD_5$, and polyphenols by 87.86%, 87.39% and 81.59%, respectively. | [100] |
| multi-soil-layering ecotechnology and adsorption on activated carbon/lime | activated carbon adsorbent, (pH = 2, T = 298 K, dilution factor = 5, mass (CA) = 5.5 g) | Reduction in COD, polyphenols and color by 92% 100% and 100%. | [101] |
| Sole and combination of $H_2O_2$, $O_3$, and UVA irradiation | Photolysis, ozonation ($O_3$/dark), ozone photolysis ($O_3$/UVA), $H_2O_2$-peroxidation ($H_2O_2$/Dark), $H_2O_2$, photoperoxidation ($H_2O_2$/UVA), peroxonation ($H_2O_2$/$O_3$/dark), and photo-peroxonation ($H_2O_2$/$O_3$/UVA) | 40% reduction in COD by UVA, increase in $BOD_5$ up to 209% and biodegradability by dark peroxonation of 254% | [102] |

The combined purification methods summarized in Table 6 were designed to remove the targeted pollutants. Obviously, the treatment of OMW using combined processes is more reliable and effective in removing certain pollutants with higher efficiency and lower cost than a single process. Integrating electrocoagulation, catalytic ozonation, and biodegradation [103] was very efficient, achieving 98.4% COD and 97.2% TOC reduction. In another research [101], integrated multi-soil-layering Eco technology and adsorption on activated carbon/lime achieved polyphenols and color by 100% and 100%, respectively. This indicates that the combined process was very successful and efficient in removing certain targeted pollutants. A COD degradation from 64 to 88% and methane generated ranged from 281 $cm^3$ to 322 $cm^3$ of $CH_4$/g COD removed was attained using a combination of Fenton's reagent and anaerobic biological process [92]. Another combination of coagulation followed by biological oxidation [1] was very efficient in reducing phenols (99.6%). On the other hand, 95.3% removal of BOD5 was achieved using a combined coagulation-flocculation system of 0.1 g of $\gamma$-$Fe_2O_3$ nanoparticles [32].

As noticed above, the main objective of the conventional treatment processes for OMW has been to remove pollutants and reduce the COD to obtain remediated water that could be used to irrigate some trees or plants. Many studies have reached high removal efficiencies of the pollutants. However, the substances were considered pollutants that should be degraded or biodegraded and removed. Accordingly, conventional treatment processes are not suitable for OMW management since the polyphenols, antioxidants, and other valuable materials are lost. For this reason, another sustainable approach is more suitable for the management of OMW to obtain the benefits of those materials [8,102]. They are more effective, environmentally friendly, and economically viable recovery materials obtained from the OMW in sustainable ways. These approaches will be covered in detail in the following sections as valorization constituents of OMW and sludge, sustainable recovery of polyphenols from OMW, and OMW economical study.

## 5. Valorization Constituents of OMW and Sludge

Sustainable utilization of wastes produced by the food-industry is becoming a hot environmental issue due to the polluting properties of the byproducts, both liquids, and solids, generated during the preparation and production of food [104,105]. For instance, considerable amounts of solid and liquid wastes are produced during olive oil extraction; these wastes are collectively known as olive mill wastes [106]. Around 20 million tons of fresh water is needed for olive oil processing in the Mediterranean area, resulting in up to 30 million tons of liquid-solid waste per year.

In the Middle East, it was reported in 2017 that around 175,000 m$^3$ of OMW were produced from 209,000 tons of olives in the olive pressing industry. This quantity of OMW roughly contains 3069, 7956, 149, 2.07, 3753, and 4.2 tons of BOD$_5$, COD, residual olive oil, phenols, total suspended solids, and phosphorous respectively. Therefore, the OMW content is high with organic matter expressed as BOD$_5$ and COD [33]. The produced OMW usually contains lipids, fats, carbohydrates, nitrogenous compounds, organic acids, polyalcohols, and some inorganic constituents. Other compounds found in OMW include a wide variety of phenolic compounds, tannins, and organic halogenated pollutants. In addition, OMW is characterized by its high suspended solids content, high turbidity, and a low pH of 3.5 to 5.5 [107].

Based on its high pollutants load, OMW is considered as a significant source of environmental pollution. Due to the high-cost obstacles, no treatment processes are currently available at the mills; therefore, OMW is usually discharged into the environment, causing significant environmental pollution such as: coloring and pollution of ground and surface waters, soil surface, and foul odors problems. However in other cases, like in Jordan, OMW is transferred in tankers to certain basins found in the northern side of the country [23].

Fortunately, many OMW constituents have high economical values and can be extracted, purified, and used in many applications. The main valuable components that can be extracted from OMW wastewater were summarized by [8] and shown in Figure 6 [8].

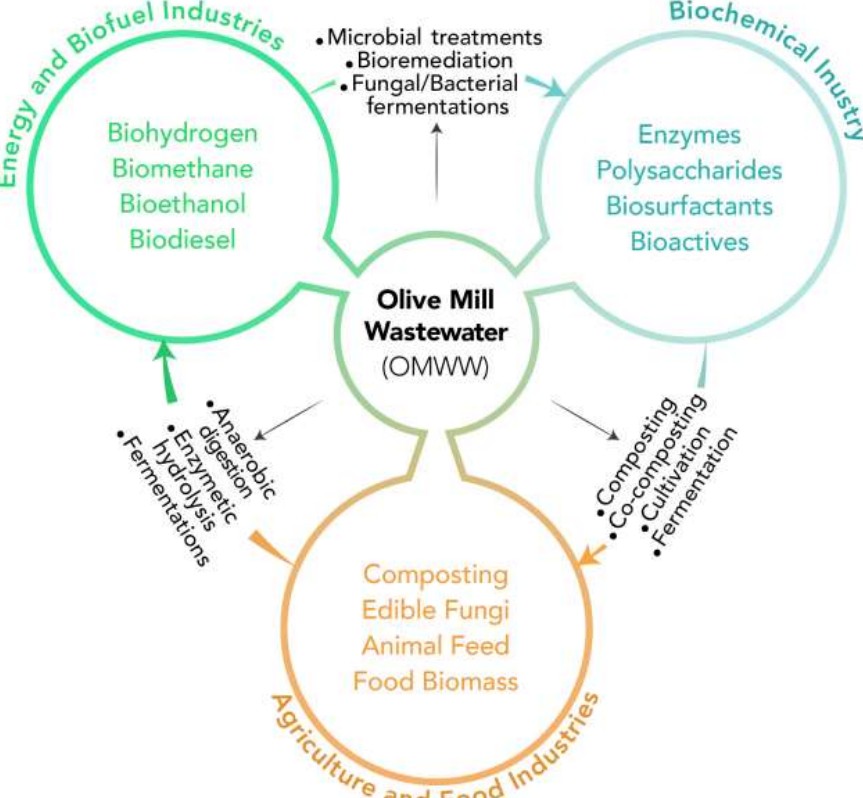

**Figure 6.** Microbial valorization and potential uses of OMW [9].

The importance and uses of these valuable materials found in OMW will be discussed in the following sections.

### 5.1. Bio-Energy Materials

The first possible process is to produce biogas from OMW. It is well known that the anaerobic methanation digestion process degrades the organic load in the wastewater and produces biogas, $CH_4$. However, for an efficient anaerobic methanation process, OMW should have a balanced "carbon to nitrogen to phosphorus (C/N/P)" ratio, a pH value of 6.5 to 7.5, and it should contain a low content of toxic materials for microorganisms. Although the C/N/P ratio in the OME is unbalanced, different researchers applied anaerobic processes using OMW as the only substrate for thermophilic bacteria [10,108]. OMW content of nutrient-rich streams, substrates, sugars, volatile acids, polyalcohols, and fats significantly enhance process performance [23]. However, for successful biological treatment, a pretreatment step should be performed to reduce the concentration of toxic materials [18].

As an example, the pretreatment of OMW by electro-coagulation, as shown in Figure 7, enhanced the performance of anaerobic digestion, which was operated at a higher organic loading rate from 4 to 7.5 g COD $L^{-1}$ $day^{-1}$ [108].

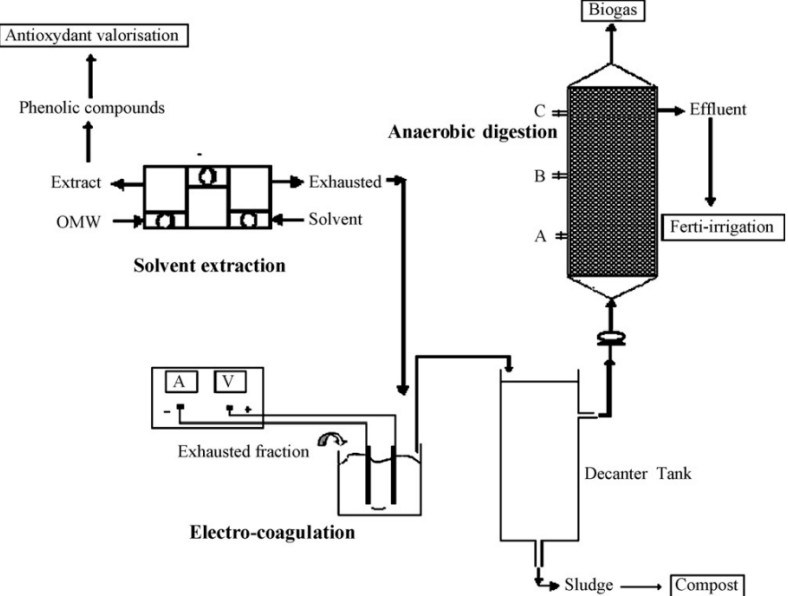

**Figure 7.** OMW treatment using solvent extraction, electro-coagulation, and anaerobic digestion. Reprinted with permission from Reference [109]. Copyright 2019. Elsevier.

Moreover, OMW with high organic concentrations was converted into methane and carbon dioxide in an aerobic or anaerobic digesters as explained by Equation (2) [110], followed by a "two-phase anaerobic digestion process" by hydrolytic and acidogenic fermentative bacteria:

$$\text{Aqueous organic load} + \text{microbes} \rightarrow CH_4 + CO_2 + NH_3 + H_2S + \text{Biomass} + \text{Heat} \qquad (2)$$

In the first phase, macromolecules such as proteins and carbohydrates are transformed into simple organic compounds such as sugars, amino acids and intermediates including alcohols, volatile organic acids, ketones, $CO_2$, and $H_2$ using hydrolytic and acidogenic fermentative bacteria. In the second phase, all these constituents are metabolized and transformed into $CH_4$ and $CO_2$ [8,111]. On the other hand, the high organic matter in OMW could be considered as a suitable source for biodiesel and ethanol production. However, it is crucial to reduce or remove its phenols in all biological processes and use its lipid and carbohydrate content to yield the required biofuels [112,113].

Yousuf et al. [114] studied the conversion of OMW into biodiesel and lipids using the yeast *Lipomyces starkeyi*. The profiles of the obtained fatty acid displayed a diffusion of oleic acid, illustrating the ability of this strain to store lipids that can be converted into biodiesel. In another study, Bellou et al. [108] used *Zygomycetes moelleri* strains to covert OMW into lipids. In this research, phenolic compounds were also separated.

*5.2. Bio-Chemical Materials*

The production of bio-compounds from OMW is useful since it produces low-cost new compounds and minimizes the environmental pollution caused by the OMW drainage [8]. Fungi is a powerful tool for OMW degradation and detoxification by removing recalcitrant compounds to produce oxidative enzymes such as phenol oxidases and polyphenol oxidases [115,116]. Table 7 shows some examples for the output of ligninolytic enzymes coupled with OMW [8].

**Table 7.** Enzymes produced by *ligninolytic fungi* during OMW biological treatment of OMW [8].

| Microorganism | Treatment Conditions | % Phenolic Reduction | Time | Enzymes |
|---|---|---|---|---|
| *Lentinus edodes* | "Five-fold diluted OMW in medium containing glucose 5 g/L". | 65 | 12 d | Phenol oxidases MnP |
| *Pleurotus ostreatus* | "10% (*v/v*) water-diluted OMW incubated with fungus previously adapted potato dextrose broth, yeast extract, and maltose medium containing up to 20% OMW" | 90 | 100 h | Phenol oxidases |
| "*Abortiporus biennis* (CCBAS 521)" | | 54.5 | | Lacc MIP |
| "*Pleurotus ostreatus* (CCBAS 472)" | OMW was 50% (*v/v*) water-diluted, pH was adjusted to 6.0 with $H_3PO_4$. The white-rot fungi cultivativation on 50% water diluted OMW plus agar 1.6% (*w/v*) | 51.5 | | MnP (only observed in |
| "*Panellus stipticus* (CCBAS 450)" | | 42.2 | | *P. ostreatus* and *A. biennis*) |
| "*Dichomitus squalens* (CCBAS 751)" | | 36.4 | 30 d | |
| "*Pleurotus flavido–alba*" | "Bioreactor filled with basal medium plus veratryl alcohol (0.43 g/L), Tween 20 (0.5 g/L), and supplemented with Mn (II). After 5 d of fermentation, concentrated and sterilized OMW was added" | 51 | 14 d | MnP Lacc |
| "*Pleurotus ostreatus*" | Undiluted, thermally processed OMW. | 65 | 21 d | MnP |
| | Thermally processed OMW at 100 °C, and water diluted at 50% (*v/v*). | 67 | 19 d | LiP |
| | 50% (*v/v*) water-diluted and sterilized, (120 °C, 1 atm) OMW | 78 | 21 d | Lacc |
| "*Pleurotus* spp. LGAM P105" | | 69 | | Lacc |
| "*Pleurotus* spp. LGAM P112" | 75% OMW and 25% distilled water, no addition of nutrients or pretreatment | 71 | 24 d | |
| "*Pleurotus* spp. LGAM P113" | | 73 | | |
| "*Pleurotus* spp. LGAM P116" | | 73 | | |

Cordova et al. [117] produced lipases from OMW fermentation based on the different amounts of the remaining oil. In another work, pectolytic enzymes were produced using OMW with the yeast strain *Cryptococcus albidus var. albidus* [118,119].

OMW is now considered an important resource for valuable biomolecules such as "exopolysaccharides (EPSs)". The use of low-cost raw compounds to yield EPSs by microorganisms has similar properties to compounds produced from vegetable biomass [120]. EPSs are polymeric materials with diverse biological functions, and their structural complexities have significantly different applications. They can be involved in the bioremediation, cosmetics, pharmaceuticals, and foods industries. Many authors investigated EPSs production from OMW [121–123]. Other "EPS-producing bacteria" that grow in OMW have also been investigated elsewhere [124–130].

Biosurfactants, considered secondary metabolites, are stable at harsh salinity, pH, and temperature and can be used in different applications such as heavy metals recovery, soil remediation, medical or food applications [131]. Moreover, OMW contents are considered a suitable carbon source to produce surfactants or biosurfactants [110,130,132,133]. Moya Ramírez et al. [134] tested the "surfactin synthesis" with *Pseudomonas* and *Bacillus subtilis* strains, using 2–10% (*v/v*) OMW solutions. It was shown that the amount of surfactin was 3.12 mg/L for *B. subtilis* in 2% OMW.

Moreover, OMW has been considered as a source of some bioactive materials such as phenolic compounds. These compounds are characterized by their antioxidant action and can be obtained from olive by-products. These antioxidants have been examined by many researchers [134,135]. Phenols exist in OMW as a colored tincture, and their concentrations differ according to their polarity, olive diversity, method of cultivation, and oil separation technologies [13,136,137]. Different high-added-value microbial products were produced based on OMW conversion, as shown in Figure 8 [14].

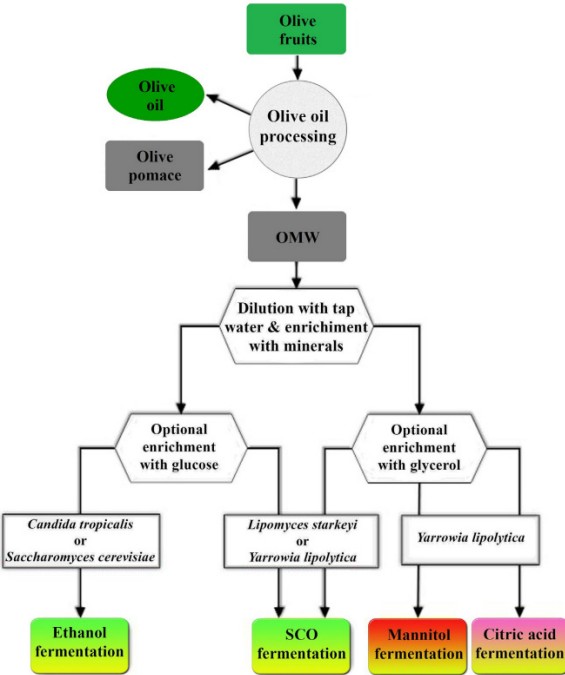

**Figure 8.** Flow chart for OMW conversion into different microbial products. Reprinted with permission from Reference [14]. Copyright 2016. Elsevier.

As mentioned above, the phenolic compounds may differ in OMW. However, the main bioactive materials illustrated in the literature are flavonoids, phenyl alcohols, secoiridoids, phenols acids, secoiridoid derivatives, flavonoids, lignans, oleuropein, Hydroxytyrosol, "M4-methylcatechol", protocatechuic acid, "4-hydroxybenzoic acid" vanillic acid, "3,4-dihydroxyphenyl glycol", homovanillic alcohol, "4-hydroxy-3,5-dimethoxy benzoic acid", "3,4-dihydroxyphenylacetic acid", "2-(3,4-dihydroxyphenyl)-1,2-ethandiol", and "2-(4-hydroxy-3-methoxy) phenyl ethanol" [138–142]. These compounds exhibit anti-inflammatory, anti-hypertensive hypocholesterolemic, hypoglycaemic properties, and

antimicrobial properties against certain fungi, bacteria, and mycoplasma [143]. On the other hand, biofilters have also been used to recover phenols from OMW [144].

Some researchers studied the preparation of some types of microorganisms using OMW to obtain different microbial biomasses. For example, in a recent study, *Edible fungi* can grow using OMW as nutrient sources [145–147]. Koutrotsios et al. [148] studied the appropriateness of water diluted-OMW as a substrate for producing *Hericium Erinaceus* biomass. The "*H. Erinaceus*" was produced in 50% OMW, yielding 0.155 g/100 mL.

### 5.3. Medicinal Constituents of OMW

A significant interest in natural antioxidants, especially phenolic compounds, due to their therapeutic and health benefits has increased during the last years. Phenolic compounds have been known for decades for their important antioxidant activity [149]. Many recent studies have confirmed this fact [150,151]. The Mediterranean countries' olive mill wastes are usually rich in valuable ingredients. The antioxidant activity of its phenolic compounds ingredients had already been tested [152,153].

Leouifoudi et al. [154] studied some Moroccan OMW extracts' antibacterial and antioxidant potentials. Their results demonstrated that OMW was rich in bio-phenols that show significant inhibition for oxidation reactions. This activity was related to the phenol's concentration and the nature of the phenolic compound's composition of the extracts. According to these results, OMW is considered a promising antioxidant source that could be used in different potential biological applications, including the biomedical domains and natural anticancer agents [154]. In the same line, Bedouhene [155] extracted polyphenols from OMW to investigate their effects on reactive oxygen species (ROS) production by a human neutrophil. Their results showed that the extracted polyphenols exert a strong antioxidant effect. They could have an anti-inflammatory effect by inhibiting neutrophil ROS production and scavenging hydrogen peroxide, thus limiting their toxic effects. These results strongly indicate that OMW could be used to extract polyphenols for medicinal applications [155].

Navarro et al. [156] studied the use of powders obtained after the ultrafiltration and nanofiltration of OMW as antiglycative agents to reduce the load of advanced glycation. Their results confirmed that the direct trapping of dicarbonyl compounds is the main route explaining the antiglycative action known as antioxidant capacity. These results support further investigations to evaluate the technical feasibility of using OMW powders as antiglycative ingredients in foods or pharmaceutical preparations in the future [156]. In a more recent study about cardiovascular toxicities as a side effect in cancer patients receiving chemotherapy, Albini et al. [157] reported that the polyphenol-rich purified extracts are a valid candidate for combination chemotherapy and cardiovascular protection from induced cardiac damage.

### 5.4. Agricultural Materials

OMW, which contains a large amount of organic matter (polysaccharides, organic acids, tannins, phenols, and lipids) and minerals, are beneficial for crops [110,158,159]. On the other hand, the solid fraction of OMW is the olive pomace, which is rich in cellulose, lignin, and phenolic compounds [160]. All these valuable compounds have been utilized in different aspects of crop production.

Land spreading of treated OMW has been implicated in enhancing agricultural soil's chemical, physical, and biological characteristics. OMW was reported to improve organic matter content and phytonutrients in soil [161,162]. Moreover, the total nitrogen content of the soil and the available potassium were increased in response to OMW [163]. Several reports indicated increased water holding capacity, increased stability of soil aggregates, enhanced soil microflora, and decreased bulk density and soil erosion due to olive oil compost (OMC) [164]. The better soil microflora in response to OMW is due to increased aerobic bacteria and soil respiration [165,166].

Treated and diluted OMWs were reported to enhance plant growth and development. This biostimulant action of OMWs is probably due to increased humic acid in treated soil and phenol biodegradation [167]. Seed germination was enhanced on medium supplemented with treated OMW, germination of durum wheat and white bean was significantly improved in response to OMW [168]. In addition, the aerobic biotreatment process has long been suggested for the treatment of OMW using different strains. For example, *Azotobacter vinelandii* microorganism can produce nitrogen to obtain a fertilizer from OMW [24,169–171].

On the other hand, OMW has been tested to yield composts in combination with "domestic sewage sludge", sawdust, cereal straws, and manures in a process called "co-composting" [172]. It was found that co-composting of OMW depends on the temperature, pH value, oxygenation moisture, nutrients, and the development of the microbial populations [173]. Typically, the best conditions for OMW co-composting process were C/N ratio of 30, moisture content of 55%, small particle size, and a suitable oxygen amount [174].

Many reviews have suggested and discussed the potential use of OMW in crop protection as a biopesticide. Such application will serve sustainable and organic agricultural production and minimize synthetic pesticides' misuse [175]. The pesticide action of OMW resulted from its antibacterial and antifungal properties. Different plant pathogens can be controlled by treated OMW, such as Pythium, Verticillium, Fusarium, and Botrytis [176].

*5.5. Animal Feed and Food Materials*

OMW has been considered a source of some biopolymers such as pectin's, cellulose, hemi-celluloses, and other products such as fat replacements and alcohol insoluble residue (AIR) [177]. Figure 9 displays the process flow for the AIR recovery from OMW. Dietary fiber consists of associated materials that are resistant to digestion by enzymes [178]. On the other hand, the presence of phenolic compounds in OMW minimizes AIR separation and requires expensive equipment and chemicals.

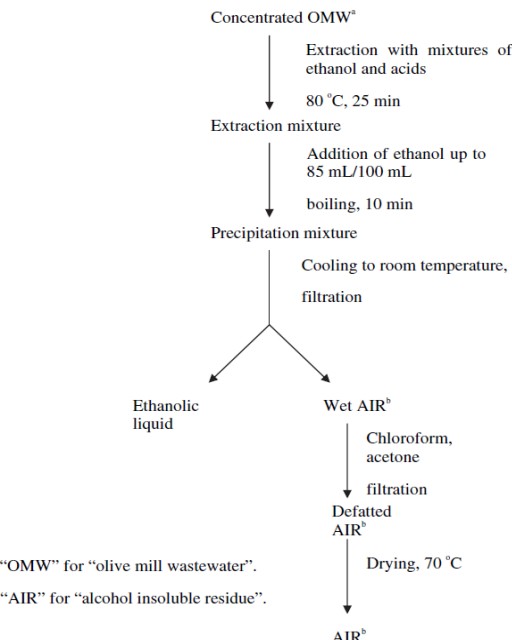

**Figure 9.** Process flow for the recovery of the AIR from OMW. Reprinted with permission from Reference [177]. Copyright 2010. Elsevier.

Some researchers used lignin-degrading fungi and solid-state fermentation to improve the nutritional properties of OMW mixed with feedstuffs (wheat middling's, barley grains, wheat bran, field beans, Shorts of wheat flour, and crimson clover) by using the *Pleurotus pulmonarius* and fungi *P. ostreatus* [110,179,180].

### 5.6. OMW Phenolic Compounds

OMW comprises high-value constituents, including phenolics, pectin, recalcitrant and essential enzymes. Its phenolic compounds result in certain toxicity/phytotoxicity. Phenols are composed of one or more hydroxyl groups (polar part) linked directly to an aromatic ring (non-polar part). They are present in plants as esters or glycosides rather than free form [181]. The chemical structures and the IUPAC names of the most abundant phenolic compounds detected in OMW are shown in Figure 10.

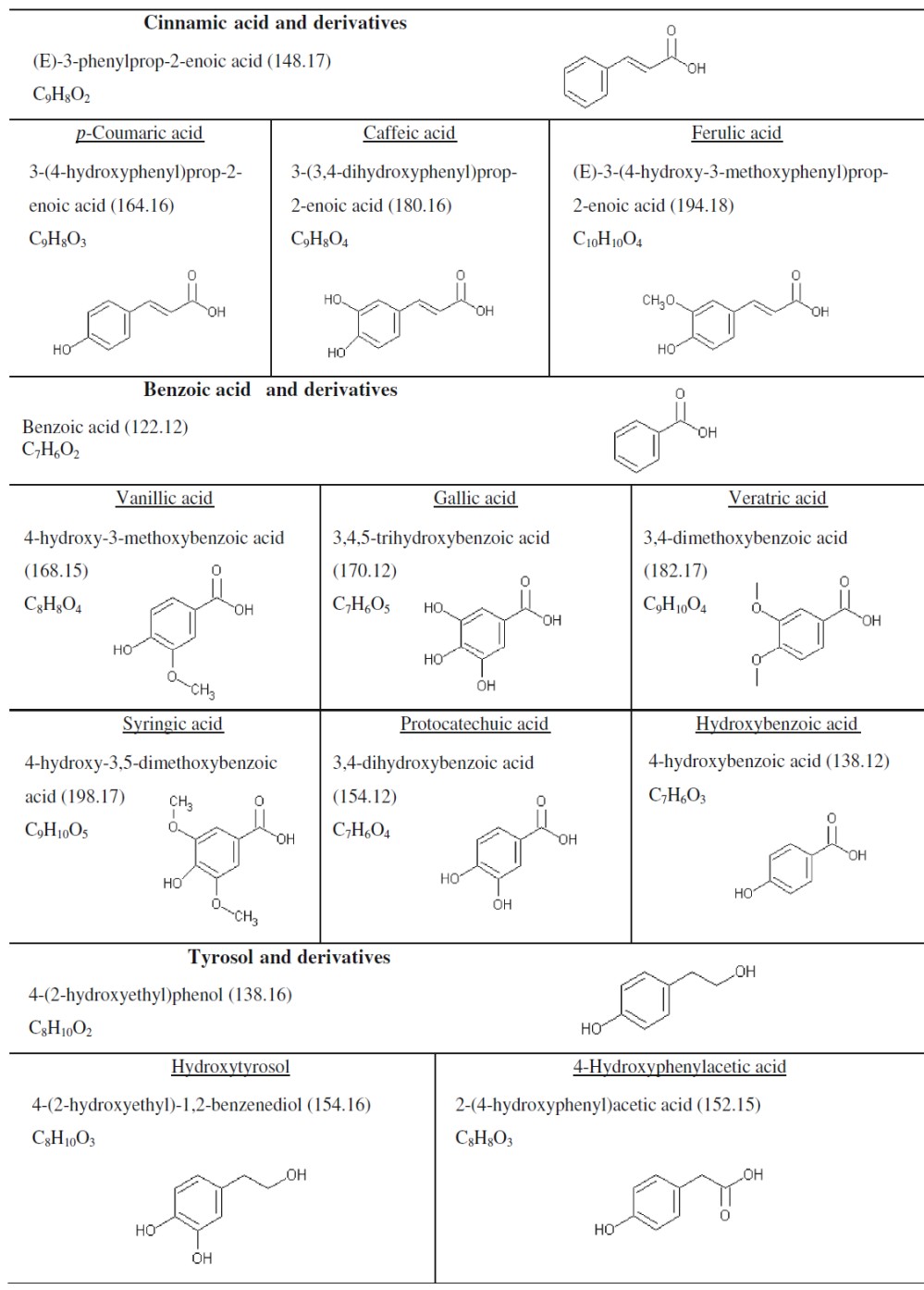

**Figure 10.** Most abundant phenolic compounds, chemical structure, and IUPAC names detected in OMW. Reprinted with permission from Reference [182]. Copyright 2011. Springer.

On the other hand, Table 8 shows the concentration ranges of the main phenolic compounds and their main bioactivities.

**Table 8.** The concentration of phenolic compounds present in OMW and their main reported activities. Adapted from References [54,138,183,184].

| Phenolic Compound | Range (mg L$^{-1}$) | Main Reported Activities |
|---|---|---|
| Hydroxytyrosol | 1073.37–4326.88 | Antioxidant, cardioprotective, Skin bleaching, antiatherogenic, anti-inflammatory, chemo preventive |
| Protocatechuic acid | 66.61–107.31 | antioxidant, antifungal, antihepatotoxic |
| 3, 4-dihydroxyphenylacetic acid | 212.26–253.42 | Antioxidant, hepatoprotective agent |
| Tyrosol | 35.53–920.16 | Antioxidant, anti-inflammatory antiatherogenic, cardioactive |
| Vanillic acid | 8.77–126.64 | antioxidant antimicrobial activity |
| Caffeic acid | 9.59–682.19 | Antioxidant, chemoprotective antiatherogenic, antimicrobial |
| Siringic acid | 1.42–24.00 | antioxidant, antimicrobial, anti-inflammatory |
| p-Cumaric acid | 29.79–341.93 | Antioxidant, antimicrobial chemoprevention |
| Ferulic acid | 23.50–27.12 | antioxidant, anti-inflammatory, |
| Oleuropein | 127.46–5093.49 | Antioxidant, hypoglycemic antihypertensive |
| Luteolin | 84.57–2745.90 | antioxidant, anti-inflammatory, antibacterial, antiviral |

As shown in Table 8, the composition of OMW varies depending on many factors such the geographic location and type of oil extraction method. Consequently, OMW phenolic compounds' concentration, type, and bioactivities usually vary according to the same factors.

It should be noted that the fraction of phenolic compounds that remains in the olive oil after its extraction from the fruit is only 2%, and the remaining 98% is lost in OMW. Moreover, Table 8 shows that most of these phenolic compounds act as antioxidants and many other bioactivities. These facts ensure that OMW is potentially a rich source of a diverse range of phenolic compounds with a wide array of bioactivities and suggest the search for suitable processes to recover these listed high fractions of phenols [185].

Phenols recovery processes involve physical processes utilized for extraction purposes, such as chromatography, supercritical extraction, and membranes applications, while biological and chemical methods are mainly used to reduce organic load [186,187]. This objective of this process is not only to recover a pure phenol (i.e., Hydroxytyrosol) but also the phenol's mixture as a crude product.

In a review paper, more than 30 phenolic compounds were found to be recovered from OMW and olive processing solid by-products [86]. In another work, membrane technology, including UF, NF, and RO membranes, was investigated to produce phenolic compounds in the retentive streams of NF and/or RO [96]. The main phenol recovery methods are summarized as follows:

## 6. Sustainable Recovery of Polyphenols from OMW

As mentioned above, OMW like many industrial wastes contain many valuable components. Fortunately, the present separation technologies for separation allow a relatively cheap recovery of these components and their reuse as functional additives. The suitable processes for separating polyphenols OMW will be discussed in the following sections.

### 6.1. Polyphenols Extraction

Among the different processes, extraction is considered a conventional method; it has been preferred in many applications for its simplicity [86]. In this process, one or more immiscible liquid components are transferred to another liquid using an extractant. Many parameters should be optimized to develop effective results of complete separation,

such as solvent nature, the volumetric ratio for solvent, OMW pH, and extraction stages number [188]. Olejniczak et al. [189] investigated the extraction of 11 different phenols and their acetyl derivatives using diethyl carbonate, hexane, and toluene were studied. Results found that diethyl carbonate was the most effective solvent and enabled high recoveries of both phenols and acetyl derivatives. This result was attributed to forming a hydrogen bond between the hydroxyl group of phenols and the oxygen atom of the carbonate group. In another study, Palma et al. [190] evaluated the batch removal of phenol from a 14.4% phenol solution in methyl isobutyl ketone using 5.5 to 6.5% NaOH solution as the extracting aqueous phase. They reported that the average removal efficiencies of phenol from the organic phase were about 97.6%, and the residual phenol in the organic layer was 1.0%. Takac and Alper [185] extracted polyphenols from OMW using ethanol up to 70% and organic acid from 0.5% to 3%.

As shown in Table 9, most of the reported studies [137,190,191] pointed to excellent phenolic extraction using an 80/20 volume mixture of methanol and water in the presence of HCl or $H_2SO_4$ as an acidic catalyst with adding n-hexane for lipids removal. Araujo et al. [192] made an elegant classification for the modes of extraction processes used for polyphenol recovery. They included many applications and the performance of each mode and concluded that the extraction of phenolic compounds depends on the chemical agent type used. Table 9 shows the modes of extraction used and some significant results found in recent studies.

**Table 9.** Methods for phenolic compounds extraction in olive pomace [193–196].

| Mode of Extraction | Solvent | Recovered Phenolic Compounds | % Rec. | Ref. |
|---|---|---|---|---|
| Solid–liquid extraction | MeOH/$H_2O$ (80:20, *v/v*, pH 2), Sodium metabisulfite n-hexane Amberlite XAD7, XAD16, IRA96 and Isolute ENV+, resins | Hydroxytyrosol glucoside; Hydroxytyrosol, Tyrosol; flavonoidal glycoside; caffeic acid; verbascoside; rutin; verbascoside isomer; oleuropein derivative; oleuropein; oleuropein isomer; luteolin. Hydroxytyrosol and others | 84 | [138,151] |
| Liquid-Liquid or Hydrolysis Liquid-solid extraction From waste cake | Ultrasonic probe and MeOH. Ethanol, a mixture of ethanol to water of 1:1 n-propanol MeOH NaOH (pH 12) HCl (pH 2) Ethyl acetate | Hydroxytyrosol and glucoside form; Tyrosol; verbascoside; verbascoside derivative; luteolin-7-*O*-glucoside; rutin; *p*-HPEA-EDA; comselogoside Hydroxytyrosol, oleuropein, luteolin, hesperidin, catechin, cyanidin glycosides and caffeic, ferulic, chlorogenic, pyrocatechinic, syringic, o-coumaric, p-coumaric, cinnamic and trans-cinnamic acids Gallic, protocatechuic; hydroxybenzoic, vanillic; caffeic; sinapic, rutin, *p*-coumaric, hesperidin, quercetin and cinnamic acids | 95.3 92.2 90.7 30 90 | [138,151] |
| Ultrasound-assisted extraction (UAE) | n-hexane ethyl acetate. EtOH/$H_2O$ (80:20, *v/v*). | Tyrosol; Hydroxytyrosol; Hydroxytyrosol acetate; luteolin; vanillin; vanillic acid; *p*-coumaric acid; caffeic acid; 3,4-DHPEA-EDA; 3,4-; DHPEA-EA; methyl 3,4-DHPEA-EA oleuropein and many derivatives | Not given | [138,151] |
| Pressurized liquid extraction | MeOH/$H_2O$ (80:20, *v/v*). | Hydroxytyrosol; verbascoside; luteolin-7-*O*-glucoside; apigenin-7-*O*-glucoside; oleuropein; luteolin; apigenin, diosmetin. | Not given | [138,151] |

It is clear from Table 9 that many high-added-value biomolecules can be extracted using different extraction modes. This extraction will reduce the phytotoxicity of these cost-effective ingredients in wastewater [151]. However, the cost of each process is not given to make a complete comparison and cost optimization that could help us choose the most recovery-efficient and cost-effective extraction.

### 6.2. Polyphenols Adsorption

The adsorption is considered a feasible, low-cost, and is the most repeatedly used method for separating phenolic compounds [197]. One study removed 95% of phenolic compounds by using sand filtration and then treatment using powdered activated carbon in a batch system [198]. While in another study, the recovery yield was less than (60%) by adsorption with Amberlite XAD16 resin and ethanol as the biocompatible desorbing phase [199].

Bertin et al. [151] proposed that Amberlite XAD7, XAD16, IRA96, and Isolate ENV+ are the four most promising adsorption resins. Results showed that the recovery of Hydroxytyrosol (77%) was achieved when no acidified ethanol was used as the desorbing phase. Ferri et al. [200] studied adsorption with IRA96 polar resin, and results showed phenol percent removal of (76%) as the non-polar adsorbents used allowed the higher desorption ratios.

Aksu and Gönen [201] studied a continuous fixed bed carried out using Mowital (R)B30H resin immobilized dried activated sludge as a bio sorbent sorption of wastewater containing a multi-component system of phenol and chromium. Results showed that the dried activated sludge's column biosorption capacity was 9.0 mg/g and 18.5 mg/g for phenol and chromium (VI), respectively. The capacity of the activated sludge column sorption for multi-component systems decreased as a result of the presence of both components. This may be due to both the components competing for the same adsorption sites on the activated sludge. Wu and Yu [202] evaluated the biosorption of various phenolic compounds from aqueous solutions by non-living *Phanerochaete chrysosporium* mycelial pellets. Results showed that with decreasing water solubility, sorption was increased.

Table 10 illustrates experimental results for different types of sorbent fitted with the Langmuir theory, where KL (L/mg) is the Langmuir constant and is representative of the affinity of the sorbate for the sorbent and $q_{max}$ (mg/g) is the maximum adsorption capacity.

**Table 10.** Langmuir constants for PCs adsorption from various absorbents are reported in the literature [203].

| Adsorbent | $q_{max}$ | $K_L$ | References |
|---|---|---|---|
| Olive pomace | 11.4 | 0.005 | [204] |
| Banana peel | 688.9 | 0.24 | [197] |
| Wheat bran | 478.3 | 0.13 | [205] |
| Activated carbon | 268.17 | 0.14 | [206] |
| Activated carbon coated with milk protein | 246.45 | 9.1 | [207] |
| PDMS/oxMWCNTs | 454.55 | 0.014 | [203] |

It should be noted that the highest values of adsorption capacity were reached with banana peel, wheat bran and PDMS/oxMWCNTs.

### 6.3. Membranes Separation Processes

Membrane processes, such as MF, UF, NF, RO, and integrated membrane processes, have been proposed in many previous works to obtain effluent streams from OMW of acceptable quality for safe disposal and the recovery, purification, and concentration of polyphenols [208]. Membrane distillation with different polymers like polyvinylidene fluoride and polytetrafluoroethylene for OMW treatment have been used [86]. The feasibility of

a Polyether–polyamide block copolymer (PEBA) membrane for the separation of aromatics was studied by Kondo and Sato [209]. The effectiveness of the membrane was tested with both synthetic and real wastewater. It was found that phenol removal for real wastewater was less due to impurities on the membrane. Phenol exhibits very low flux across the membrane as a result of the low vapor pressure of 0.055 kPa at 250 C. This was enhanced by Hao, Pritzker, and Feng [210] by making the membrane perm selective to the phenol and increasing the removal rate. Non-porous membrane (dense) has very low flux across the membrane, and this was overcome by Das et al. [211] by having polyurethane–polyacrylate modified porous membrane. The feasibility of poly (dimethyl siloxane), PDMS, and poly (methyl vinyl) siloxane (PVMS) in the membrane extraction of water–phenol mixtures with a sodium hydroxide solution was tested by Xiao et al. [212] PDMS was found to be felicitous for the operation. The influences of parameters such as feed flow rate, phenol concentration, and temperature of the process were studied.

El Abbassi et al. [208] used the direct contact membrane distillation (DCMD) process using polytetrafluoroethylene (PTFE) membranes to separate polyphenols from OMW with 100% removal of phenolic compounds. In another study, DCMD using sand filtration and subsequent treatment with activated carbon in a batch system achieved 95% removal of phenolic compounds [198].

Paraskeva et al. [213] produced OMW by a three-phase system using a process set-up consisting of UF, NF, and RO membranes. The authors studied many operational parameters to obtain the optimum operation of membranes. In order to reduce membrane fouling, a pre-treatment step of OMW filtering was conducted. Results showed that the UF process effectively separated the constituents of high molecular weight and any suspended solid particles or aggregates.

The NF and RO processes effectively separated the most significant part of polyphenols contained in the feed OMW. The "toxic" fractions with the potential for use as growth inhibitors of some native plants were accumulated in the concentrate fractions. Total phenolic content (TPC) recovery from larger waste volumes was studied in pilot-scale experiments using UF, NF, and RO membranes pilot plant illustrated in Figure 11 [96]. TPC concentration values measured at all stages of the experiment in pilot-scale at all fractions of the concentrate and permeate streams are summarized in Figure 11.

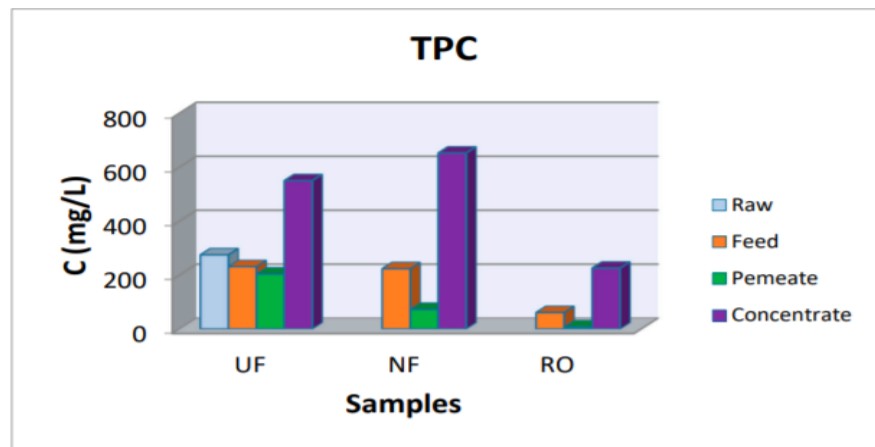

**Figure 11.** TPC concentration at the raw, feed, permeate and concentrate streams at the UF, NF, and RO membranes [96].

Figure 11 displays obtained TPC experimental data; at the UF concentrate stream, a significant amount of TPC is recovered with ~550 mg/L, while at the NF membrane concentrate stream, about 652 mg/L of TPC was retained. However, in RO concentrate stream, only 225 mg/L was recovered. The corresponding values for the permeate stream of UF, NF, and RO were 203, 37, and 7 mg/L, respectively.

The recent developments made in the management of OMW were analyzed [96]. They found that in the last 20 years, there has been a paradigm shift from simple detoxification of OMW to its valorization, based on used treatment strategies described by Figure 12.

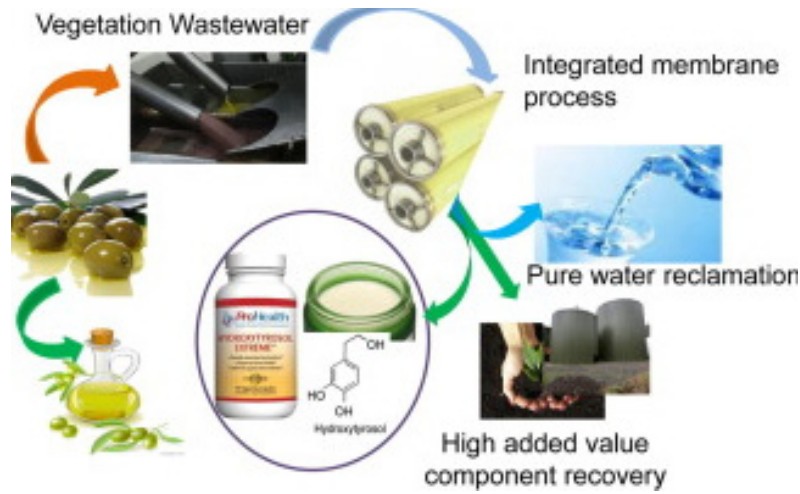

**Figure 12.** New trends in OMW valorization. Modified from [99].

A summary of membrane processes used to recover OMW valuable materials is shown in Table 11. Reprinted with permission from Ref. [99]. Copyright 2016. Elsvier.

**Table 11.** Membrane technology applied to recover OMW valuable materials.

| Type of Membrane Process | Main Results | Ref. |
|---|---|---|
| Direct contact membrane distillation using polytetrafluoroethylene membranes | 100% phenol separation | [208] |
| UF with regenerated cellulose membranes/NF/RO | 0.5 and 30 g/L total polyphenols concentrated | [214] |
| Liquid membrane with 2% Cyanex 923 | 90–97% phenol was rejected | [215] |
| Micellar enhanced UF/anionic surfactant (sodium dodecyl sulfate salt, SDS)/hydrophobic polyvinylidene fluoride membrane | 74% polyphenols were rejected | [216] |
| NF/MF/osmoticdistillation/vacuum membrane distillation | Recovery of 78% of the initial content of polyphenols | [217] |
| MF/UF/RO consists of a polymeric hydraulics membrane | Retentate containing 464.870 mg/L with free low MW polyphenols | [218] |
| Oil removal and TSS settling/UF/permeate mix with sewage and double-stage biological treatment | 50% COD load decrease after UF and up to 70% after biological treatment | [219] |
| UF/treatment with adsorbing polymers and RO | COD reduction (UF) up to 63%, 93% (RO) and 99% total | [220] |
| Centrifugation and UF of centrifuge supernatant | 55% COD, 80% ashes, and TSS reductions after centrifugation, 90% final COD abatement | [221] |
| pretreatment among flocculation/UV-TiO$_2$ photocatalysis/aerobic digestion/MF, followed by UF + NF + RO | Overall COD abatement 98.8–99.4% | [222–224] |
| pH adjustment/cartridge filtration and UF | COD, TOC and SS removal ratios 92.3%, 92.7%, and 97.1% | [225] |
| Centrifugation/UF/NF and RO | COD removal 59.4–79.2% for NF, whereas 96.2–96.3% for RO | [226] |
| MF, NF, and OD or VMD | MF, 91% and 26% for TSS and TOC reduction, respectively. NF removed 63% TOC, and TC reduction in MF permeate | [217] |

It is clear from Table 11 that membrane recovery of OMW valuable ingredients is an efficient process. The average percentage recovery is about 90% in most of the literature results.

## 7. OMW Economical Separating Phenolic Compounds Study

OMW phenols have high antioxidant potential and are considered essential additives for different consumable products [227,228] and cosmetic applications [229,230]. The main interest in OMW phenols presents an opportunity for high-added-value products that can cover the treatment cost of the waste and show a significant margin for profit. Several authors tried to calculate the economics of phenol extraction. Pan et al. [231] examined the adsorption of 4-nitrophenol on a hyper-cross-linked polymeric resin from amberlite XAD-4. The authors tested the cross-linked resin for 25 adsorption/desorption cycles with no significant changes. An estimated 20–50 cycles of resin lifespan would lead to a production cost of 3.7 to 1.6 EUR/g of phenolic compounds separated from the process.

Frascari et al. [232] studied the cost of OMW phenol separation through resin adsorption after microfiltration, a separation cost of EUR 1.7–13.5/kg of phenolic compounds was calculated, but with an assumed resin lifespan of 500 cycles, with the authors reporting that this number should be confirmed through further investigations. They concluded that applying adsorption/desorption technology for phenolic compounds found in OMW can be successfully integrated with a post-anaerobic treatment step. This combined system represents a promising solution that compromises both valorization and treatment of OMW.

Recently, Innocenzi et al. [233] investigated two case studies to recover polyphenols and water. In the first method, as shown in Figure 13.

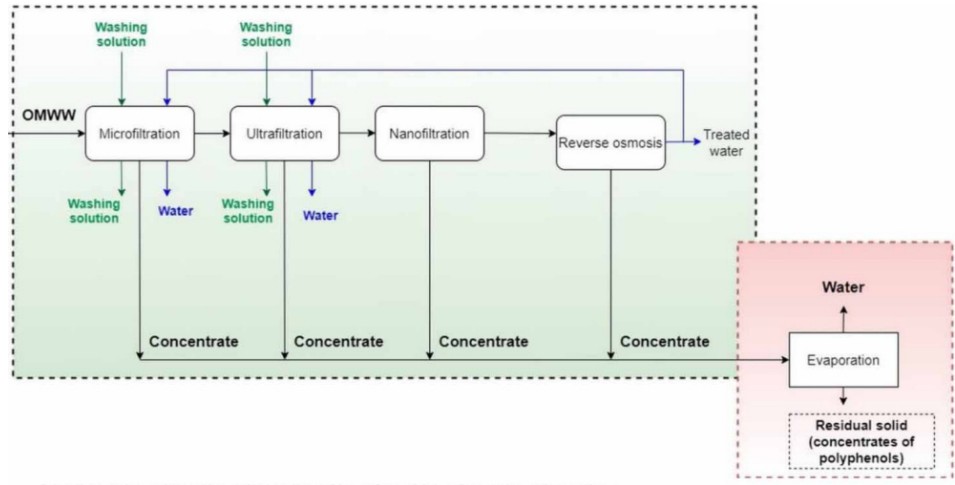

**Figure 13.** Treatment of OMW is free from pesticides [233].

As shown in Figure 13, membranes series processes were suggested to remove pollutants and concentrate the polyphenols for potential reuse. The process consists of MF membrane to separate oleic acid mainly and the partly polyphenols, then the permeate goes to the UF membrane to separate polyphenols and glucose; after that, the permeate is sent to an NF, to remove the residual polyphenols to produce a permeate containing mainly water and minerals.

The second method is shown in Figure 14, wherein the OMW proposed contains pesticides.

A specific removal process is suggested; MF, UF, and NF membranes. Then, the permeate passed to the Fenton advanced oxidation process to remove pesticides and polyphenols. The authors investigated the economic feasibility of both case studies by simulation using life cycle cost analysis (LCC) (ISO 14040, 2006). Considering the mass and energy balances obtained from process analysis using equipment and raw material purchase as primary item costs used for analysis. The total treatment cost (EUR/m$^3$ of OMW) in the first study was 253 EUR/m$^3$ of OMW and 292 EUR/m$^3$ of OMW in the second study [232].

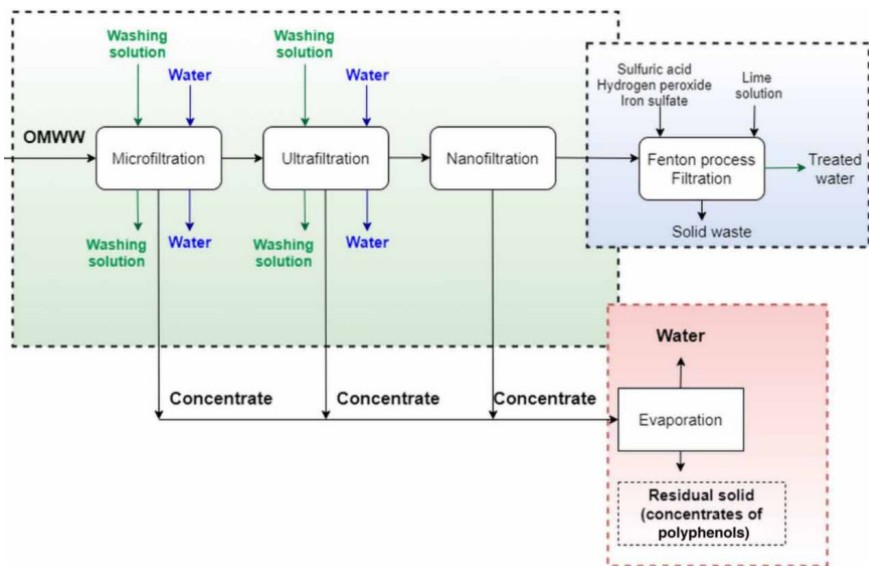

**Figure 14.** Treatment of OMW that contains pesticides [233].

Zagklis et al. [234–236] developed a phenol combined separation treatment system in a series of studies, as shown in Figure 15. That design aimed to identify all economic parameters, including the operational cost. Then they examined the margin of profit that could be achieved based on the high prices of phenols.

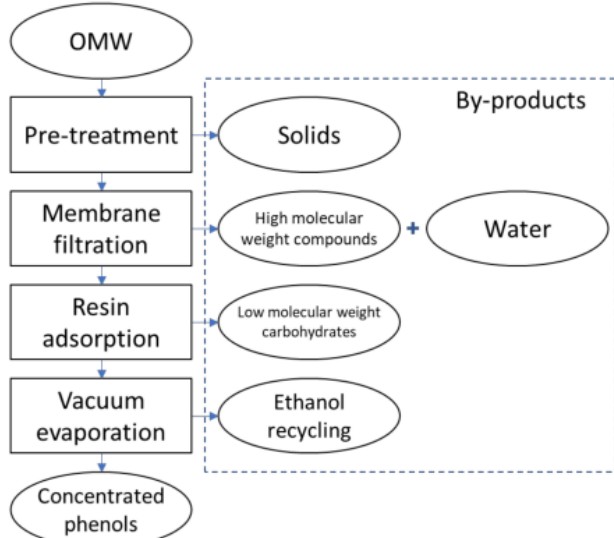

**Figure 15.** The combined treatment system for the recovery of phenols from OMW [235].

As shown in Figure 15 the proposed treatment system compromises filtration through membranes, adsorption/desorption on resins, and vacuum evaporation. According to the results of Zagklis et al. [235], the same system can be applied to the solid waste by adding an extraction step. The materials tested were OMW, leaves, and grape marc. They found that the separation cost of phenols ranged from 0.84 to 13.6 EUR/g of phenols for a resin lifespan of 5–100 adsorption/desorption cycles. They concluded that their treatment system could potentially cover the treatment cost of the waste and could make a significant profit [235].

According to Innocenzi et al. [233], the price of polyphenols is varied according to the product quality; consequently, an economic analysis taking into account different selling prices ranging from 1 to 10 EUR/kg, was made, analysis results are shown in Figure 16.

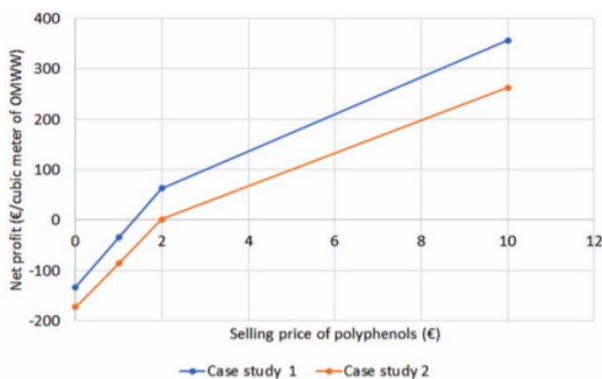

**Figure 16.** Net profit as a function of selling price for polyphenols [232].

It is clear from Figure 16 that the breakeven point was around 1.5 and 2 EUR/kg for both study 1 and 2 shown in Figures 13 and 14, respectively. This means that the first process is more economically feasible than the second when assuming the selling price of 2 EUR/kg for polyphenols. The price range calculated for the product of the process is very promising as a result of the typical value of antioxidants and the low concentration of phenols needed for food supplements and cosmetics. However, these processes in the present days of COVID-19 days are not stable due to the significant variation in the prices of petrol and shipping.

**8. Circular Economy**

Olive oil production is related to residues, olive pomace, oil and dark brown waters production, which are considered to be toxic compositions that could damage the ecosystems if discharged to the environment [23,236]. The high amount of oils and fats, polyphenols and total solids (please see Table 3) requires effective treatments in order to improve the sustainability of the sector [237].

Circular economy (C.E.) is considered one of the winning studied solutions to handle OMW production; the residues are either reused or integrated into a new production cycle. C.E., was earlier conceived by Stahel and Reday (1976) by focusing on industrial economics. They explained the industrial strategies importantly for waste prevention [238].

In recent years, a large number of scientific works has emerged with the aim of understanding C.E. models and tools, especially within the agri-food area, like the olive oil supply chain [239,240]. As the economic value of olive oil waste and by-product highlights, such a resource can be recycled and reused, thus minimizing waste [241,242]. A biochemical analysis of OMWs residues was performed to characterize the composition and properties of the starting biomass output(s)—which is an important step in choosing a beneficial conversion technology pathway [237,243].

Several conversion technologies are currently used for different residues generated during olive oil production by the industry, as shown in Figure 17 [244], which illustrates the valorization scheme of the low/medium residues (olive trees, leaves, stones) and high moisture (olive mill wastewater) residues.

The conversion of low/medium biomasses be directed to heat and power production by thermochemical processes or be upgraded into added-value products by biochemical/chemical routes (involving fractionation and hydrolysis of polymers into oligosaccharides or monomers). The conversion of wet biomass (wastewater streams) generally involves bio fertilization and anaerobic digestion industry [243].

However, many conversion technology pathways are still under investigation for future feasibility productions. Stempfle et al. [237] reviewed the available literature about circular economy pathways, as shown in Table 12.

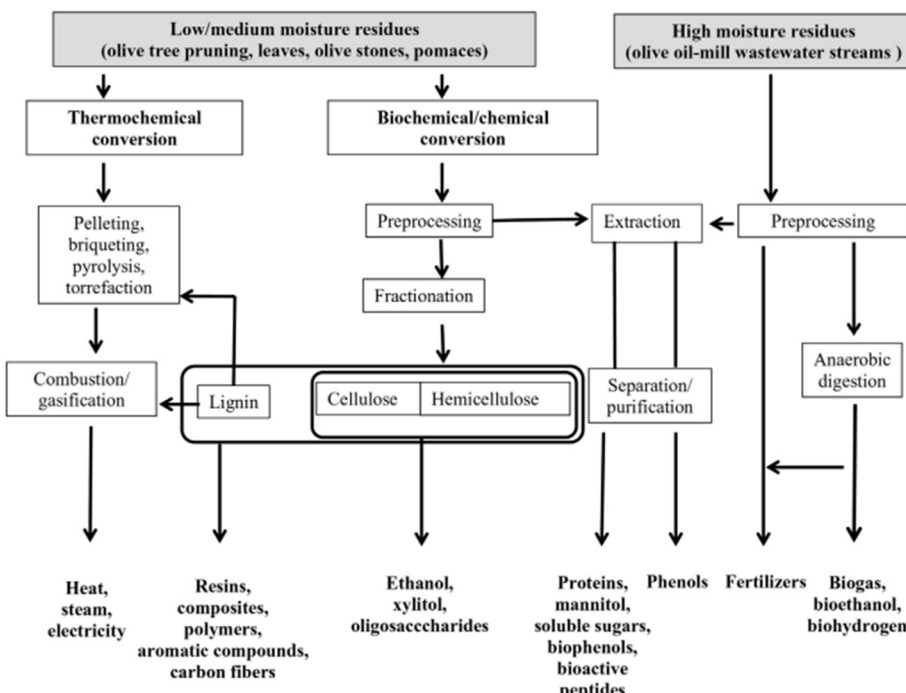

**Figure 17.** Valorization Scheme for the Different Residues Generated in the Olive Oil Industry. Reprinted with permission from Reference [234]. Copyright 2015. Elsevier.

**Table 12.** Different identified circular pathways and the related literature [237].

| ID | Name of the Pathway | No. of Total Studies | Some Studies Related to Each Pathway |
|---|---|---|---|
| #1 | High-added value bioactive compounds recovery from olive mill waste, olive leaves or waste cooking oil | 24 | [245–252] |
| #2 | Biofuel production from pruning residues and/or olive mill wastes, or waste cooking oil | 23 | [252–259] |
| #3 | Olive mill waste reused as component in the manufacture of sustainable building materials | 8 | [260–263] |
| #4 | Olive mill wastewater reused for soil conditioning/fertilization/irrigation | 6 | [264–266] |
| #5 | Pruning residues and/or olive mill waste valorized for regenerative agriculture | 6 | [267–269] |
| #6 | Biochar (bio-oil, syngas) production from olive mill waste and/or pruning residues | 6 | [270–272] |
| #7 | Olive leaves or olive cake reused for animal feed | 4 | [273,274] |
| #8 | Polymeric biomaterials production from pruning residues, olive mill by-products or waste cooking oil | 4 | [275,276] |
| #9 | Olive mill waste recycled as bio-adsorbent material for treating aqueous effluents | 3 | [277,278] |
| #10 | Biofertilizers or bio-stimulants and biopesticides production from olive mill waste | 3 | [279] |
| #11 | Treated urban/industrial wastewater reused for agricultural purposes | 2 | [280] |
| n.a | Miscellanea: collection of studies that are not focused on a specific pathway | 12 | [281–284] |

As shown in Table 12, the reviewer has addressed eleven pathways. The most investigated pathways with the largest number of published studies are: (1) recovery of high-value bioactive compounds from OMW, olive leaves, or waste cooking oil [245–252] Thus, the largest recoverable molecules from OMW are comprised of polyphenols, most of them found in olive mill wastewaters or olive pomace, and other small amounts of components like tocopherols and phytosterols. Extraction mainly occurs by using physicochemical procedures or by using the new, more sustainable techniques based on membrane technologies, such as ultrafiltration, microfiltration, nanofiltration [209], or reverse osmosis; the considerable market value of the extractable materials makes the application of such new promising value chains potential leverage for enhancing milling industry economic sustainability; the second (2) studied literature related to the production of biofuel from pruning residues of olive mill wastes, or waste cooking oil [252–259]: in this conversion pathway, the researcher concentrated on the utilization of olive tree clipping biomass, olive oil consumption wastes and olive oil industry by-products for energy production (heat, electricity, or biofuel), that presently act as the best-established valorization choices. The energy produced included: syngas, electric energy, biogas, biofuel, and combustible products from the olive pomace, where the most suggested and ready technology is gasification. The third literature was (3) reuse of olive mill waste as a component in the sustainable building materials manufacture [260–263]: to industrialize greener building materials for the construction industry.

Based on the above discussion, a comparison can be made between the conventional and sustainable treatment processes as shown in Table 13.

**Table 13.** Comparison between conventional and sustainable treatment processes applied for OMW.

| Process \ Parameter | The Conventional OMW Treatment Process | Sustainable OMW Treatment Process |
|---|---|---|
| Sustainability | Less sustainable | More sustainable |
| Environmental effect | Less friendly | More friendly |
| Recovery of valuable compounds | None to low | High |
| Energy consumption | High | Yes because of the possible use of renewable energy (Solar energy) |
| Products economic value | low | High due to the valuable products |
| Production of waste Materials | high due to different types of solid wastes produced | Low |
| Use of new separation technologies | Rare | Very necessary |

As shown in Table 13, sustainable treatment processes are more favorable than the conventional processes due to many factors including the production of very valuable products such as polyphenols, the production of less solid wastes, and the use of more new separation technologies to obtain pure products. This necessitates the application of sustainable treatment processes, rather than the conventional processes in all countries.

## 9. Concluding Remarks and Recommendations

As mentioned above, OMW is a very concentrated wastewater containing a high pollution load with a high content of toxic materials. Many reviews have been published discussing treatment, disposal and management alternatives of OMW [169] and the recent research studies employing conventional treatment processes focus on their efficiency to reduce OMW toxicity of OMW [285,286]. Other reviews focused on the farming of olive trees, manufacturing of olive oil, packaging, transportation and reverse logistics work. Accordingly, this review is the first that focuses on a recent trend in OMW management,

which compromises both valorizations of its constituents and treatment of the residual wastewater. To reach this focus point, it was necessary to discuss and analyze some critical issues concerning OMW, such as its constituents, classical treatment processes, nature and benefits of its constituents, and the performance of the main processes used to recover them. In addition, a comparison between the cost of both the recovery of valuable constituents and the treatment of residual water with the profits obtained by selling the recovered materials. Based on the above results and discussion, the following concluding remarks and recommendations are presented to help orient future research activities.

- OMW is very complex wastewater containing toxic but valuable constituents that are of vital importance. This fact suggests the application of sustainable processes that can recover the valuable constituents from fresh OMW, treat the residual wastewater and reuse the final treated water.
- Unfortunately, only a few published research papers have followed this approach (Innocenzi et al. [233] and Zagklis et al. [235]). Accordingly, intensive research should be carried out to investigate the most suitable processes for both the recovery and treatment steps.
- The required recovery processes should be mild and passive towards the chemical structure of the constituents, such as antioxidants and others. In other words, the recovery method should not be accompanied by any change in the chemical structure of the chemicals, which must not to lose their properties.
- According to the literature survey below, it seems that most research on OMW is from Mediterranean countries. However, such research is of a separate and fragmented nature, with little or no actual collaboration between researchers from different countries. This suggests the urgent need for donors to provide grants for relatively large projects that join researchers from different countries to achieve more successful and substantial results.
- Among the recovery methods, liquid–liquid or liquid–solid extraction using suitable selective solvents such as ethyl acetate and adsorption using selective adsorbent show a high percentage recovery of phenolic compounds from OMW. More research is needed to optimize these lab or pilot plant scale processes.
- Among the numerous research papers on this topic, only a few consider scaling up their experiment, such as Zagklis et al. [236]. More research should consider the large-scale application of single or combined recovery and treatment systems.
- One of the problems that usually faces OMW is that it is a seasonal phenomenon. This means that it appears in a specific period of the year that extends from October to January or February. This means that the fresh OMW is available only in this period, which puts pressure on the experiment's teams. This fact encourages researchers to find a suitable OMW storage method that keeps it fresh with negligible degradation of the valuable constituents.
- More research could be performed to obtain the polyphenols and other valuable compounds from olive leaves.

**Author Contributions:** Conceptualization, Z.A.-Q.; literature search, designing of figures and tables, writing—original draft preparation, Z.A.-Q., H.A.-Z., B.H., W.O. and S.A.-R.; validation and supervision, Z.A.-Q.; writing—review and editing, Z.A.-Q.; provided comments and proofread, M.S. and Z.F. All authors have read and agreed to the published version of the manuscript.

**Funding:** This research received no external funding.

**Institutional Review Board Statement:** This study did not involve humans or animals.

**Informed Consent Statement:** This study did not involve humans.

**Data Availability Statement:** This study did not report any data.

**Acknowledgments:** The authors thank the administration of library of Al-Balqa applied university, Jordan, for their help in obtaining all references full text.

**Conflicts of Interest:** The authors declare no conflict of interest.

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
