# Peer review of "Sustainable vs. Conventional Approach for Olive Oil Wastewater Management: A Review of the State of the Art"

_water, doi:10.3390/w14111695_

Round 1
Reviewer 1 Report
Comments to author(s)
The manuscript deals with the topic of olive mill wastewater (OMW) management, which is of great interest for scholars, economic actors and policymakers around all the Mediterranean Region, due to organizational and environmental problems related to the huge annual production rate of such effluent, and to its high load of pollutants with severe phytotoxic effects.
However, the manuscript shows a merely descriptive cutting, and does not discuss critically the findings achieved. The presented results are not very consistent with the declared purpose of the research, and the significance of the work is not evident.
Overall, the fluency of the text is compromised by many redundancies, and by the detailed technicalities that do not serve the cause of discussing two different approaches to OMW management, while making the manuscript quite hard to read. In addition, the English should be checked for grammar and style by a proof-reader.
Major concerns:
- Methodological framework
The authors presented a manuscript based on literature review, but the methodological framework is not described at all. According to academic standards, and to the Journal’s instructions for authors, a section reporting with sufficient detail the methods and procedures employed must be added. Authors should explain which kind of review has been carried out (i.e., systematic, scoping, …?), which search strategy guided the identification of relevant studies, which literature sources have been consulted, basing on which eligibility criteria the studies have been selected for the review, the total number of the selected studies, how data were charted and analysed. I suggest following appropriate guidance on the reporting of reviews, such as the PRISMA protocol.
- Definition of conventional VS sustainable OMW treatment processes
The conventional or sustainable nature of OWM treatment processes seems to be taken for granted, but it is not explicated on the basis of what parameters or criteria such distinction was made. Sustainable processes seem to be associated with the valorization of OMW constituents (section 5) and the recovery of polyphenols from OMW (section 6), but it is not clear if these processes are more sustainable from an environmental perspective (less impactful, low-input, more eco-friendly processes, etc.), or from a resource-efficiency point of view, given the opportunity of keeping longer into the production and economic cycle materials with “many application” (line 525) and “high economical values” (line 524).
- No references to circular economy
In relation to the last point, it is noticeable that the theme of circular economy (or circular bioeconomy) is completely stranger to the manuscript. I suggest that the focus of the work should at least consider and refer to the circular approach, if not be re-framed into such key-paradigm, when approaching sustainability. Please find some references about
- Circular economy as a sustainability paradigm:
- Geissdoerfer, M.; Savaget, P.; Bocken, N.M.; Hultink, E.J. The Circular Economy—A new sustainability paradigm? Journal of Cleaner production 2017, 143, 757–768. https://doi.org/10.1016/j.jclepro.2016.12.048
- Circular economy applied to the olive oil supply chain:
- Donner, M.; Radić, I. Innovative Circular Business Models in the Olive Oil Sector for Sustainable Mediterranean Agrifood Systems. Sustainability 2021, 13, 2588. https://doi.org/10.3390/su13052588
- Stempfle, S., Carlucci, D., de Gennaro, B.C., Roselli, L., Giannoccaro, G. Available pathways for operationalizing circular economy into the olive oil supply chain: Mapping evidence from a scoping literature review. Sustainability 2021, 13(17), 9789. https://doi.org/10.3390/su13179789
Economic analysis
In the introduction, authors state that “these processes will be economically analyzed and compared their benefits” (line 55), but section 7. (“OMW economical study”) only reports some data from the reviewed literature about the estimated costs of separating phenolic compounds from OMW with different technologies, and the estimated net profits as a function of polyphenols selling price. Economic analysis has not been sufficiently addressed, and any comparison have been performed between the benefits of applying “conventional” VS “sustainable” OMW treatment processes.
Discussion
The results of the review are reported in a very detailed way at a descriptive level, presenting carefully the available techniques and procedures for treating OWM, and the possible applicative destinations of its separated valuable constituents. However, the critical discussion is very poor and should be further developed. In particular, according to the declared aims of the research, a comparison between the two explored approaches for managing OMW should be deeper argued, and innovative elements of the proposed integrated-hybrid approach should be better pinpointed.
Other minor concerns:
- For corroborating what said in the 4th bullet point of section 8. (“Concluding remarks and recommendations”), some elements of descriptive analysis of the explored body of literature should be reported
Author Response
Dear Editor
Our response to reviewer is attached

Reviewer 2 Report
Thank you for inviting me to review the manuscript titled “Sustainable vs Conventional Approach for Olive Oil Wastewater Management: A Review of the State of the Art”. The researchers collected and analyzed the recently published articles regarding the conventional and sustainable treatment processes for olive mills wastewater (OMW). The conventional treatment processes concern the environmental regulations, but the sustainable scenarios aim also to recover the valuable constituents. Although the study objective is clear, some points should be considered before final acceptance:
- Do not repeat the words in the title to the “keywords”.
- The last paragraph of Introduction should include the study objectives/procedures in brief.
- The English in the present manuscript requires improvement. Please carefully proof-read spell check to eliminate grammatical errors.
- In “Table 5. Summary of oxidation techniques for OMW treatment” and “Table 6. A summary of some recent combined treatment methods and their achieved level of purification”, include a column for the operational condition of each case.
- The subscript and superscript letters should be revised throughout the paper, such as “Fe2O3” in Line 489.
- The authors should reveal the difference between this article’s objectives and those reported in the literature such as:
A Review of Waste Management Options in Olive Oil Production
https://doi.org/10.1080/10643380490279932
Technologies for olive mill wastewater (OMW) treatment: a review
https://doi.org/10.1002/jctb.1553
Olive oil mill wastewaters before and after treatment: a critical review from the ecotoxicological point of view
https://doi.org/10.1007/s10646-011-0806-y
Environmental impacts in the life cycle of olive oil: a literature review
https://doi.org/10.1002/jsfa.8143
Author Response
Dear Editor
Our Response to reviewer 2 is attached

Round 2
Reviewer 1 Report
The paper has been revised, but there are still weaknesses, mainly regarding the methodology. The Authors don't explain which kind of review has
been carried out (systematic or scoping?), which search strategy guided the
identification of relevant studies, which literature sources have been consulted (scopus database, web of science ? others ?)
which eligibility criteria have been used, the total number of
the selected studies. I renew my suggestion to follow appropriate guidance on the reporting of reviews, such as the PRISMA protocol.
Please insert title for table 13 and review line 1 and the line regarding Energy consumption : YES ???
I don't find an improvement of English language in the text
Reviewer 2 Report
The manuscript has been improved.
